# Plasma proteomics for biomarker discovery in childhood tuberculosis

Andrea Fossati[1,2,3,21], Peter Wambi[4,21], Devan Jaganath[5,6,21], Roger Calderon [7], Robert Castro[5,8], Alexander Mohapatra [5,9], Justin McKetney [1,2,3], Juaneta Luiz[10,11], Rutuja Nerurkar[5,6], Esin Nkereuwem [12], Molly F. Franke [13], Zaynab Mousavian[14,20], Jeffrey M. Collins[15], George B. Sigal [16], Mark R. Segal[17], Beate Kampman [12,18], Eric Wobudeya[4], Adithya Cattamanchi[5,19], Joel D. Ernst [5,9], Heather J. Zar[10] ✉ & Danielle L. Swaney [1,2,3] ✉ On behalf of the COMBO Study Consortium*

Failure to rapidly diagnose tuberculosis disease (TB) and initiate treatment is a driving factor of TB as a leading cause of death in children. Current TB diagnostic assays have poor performance in children, thus a global priority is the identification of novel non-sputum-based TB biomarkers. Here we use high-throughput proteomics to measure the plasma proteome for 511 children, with and without HIV, and across 4 countries, to distinguish TB status using standardized definitions. By employing a machine learning approach, we derive four parsimonious biosignatures encompassing 3 to 6 proteins that achieve AUCs of 0.87–0.88 and which all reach the minimum WHO target product profile accuracy thresholds for a TB screening test. This work provides insights into the unique host response in pediatric TB disease, as well as a non-sputum biosignature that could reduce delays in TB diagnosis and improve the detection and management of TB in children worldwide.

Tuberculosis (TB) is the leading cause of mortality from an infectious disease worldwide, with 10.8 million cases and 1.3 million deaths each year[1]. Children suffer a disproportionate burden: 12% of TB disease occurs in children, but children account for 15% of TB deaths[1]. This disparity is largely due to delays in diagnosis and proper treatment initiation, as 96% of deaths are in children for whom treatment had not been initiated[2]. While sputum-based diagnostic testing is routinely performed in adults, children are unable to reliably expectorate sputum, and sputum induction is typically required. Moreover, microbiological testing has sub-optimal sensitivity due to paucibacillary disease with low bacterial burden in children[3]. As a consequence, there is a large case detection gap where an estimated half of the children with TB disease, and two-thirds of those less than 5 years old, are not reported to public health programs[1]. Consequently, the development of non-sputum biomarker TB tests is a global priority to improve TB diagnosis in children.

The majority of TB biomarker discovery studies have been done in adults[4]. In particular, host plasma protein biosignatures have shown promise for TB screening in adults, and have the potential to be translated into a simple point-of-care test[5–8]. Unfortunately, these adult biomarkers have not been validated in children[6] and translate poorly to pediatric TB disease due to different immune responses and disease manifestations in children[9]. A systematic review found that while there were pediatric-specific blood-based host markers that could meet the WHO target product profile for a TB screening test (≥ 90% sensitivity and ≥ 70% specificity)[10], there was wide heterogeneity, with the majority being from lower quality case-control studies with unclear reference standards[11], and overall requiring further validation. Thus, limited biomarker candidates for childhood TB currently exist and the development of robust pediatric-specific host biosignatures is a global priority for early detection of pediatric TB cases[12].

A full list of affiliations appears at the end of the paper. *A list of authors and their affiliations appears at the end of the paper. ✉ e-mail: heather.zar@uct.ac.za; danielle.swaney@ucsf.edu

While mass spectrometry (MS) based proteomic analysis enables a broad untargeted approach to biomarker discovery, previous plasma proteomics efforts to identify plasma biosignatures of TB disease have been limited by small sample size, variable reference standards, and exclusive use of healthy controls that overestimate performance by selection of general inflammation markers rather than TB-specific markers[7,13]. Past studies also frequently utilized samples from a single region, leading to the discovery of candidate biomarkers that may reflect the co-morbidities and environment specific to the setting, and that fail to validate elsewhere. In this work, we utilize high-throughput plasma proteomics and well-characterized pediatric TB cohorts across four countries to derive a host-based biomarker signature that differentiates childhood TB disease from other causes of respiratory disease.

## Results

### Clinical characteristics of the cohort

We included plasma samples from 511 children with presumptive pulmonary TB from The Gambia ($n = 120$), Peru ($n = 100$), South Africa ($n = 111$), and Uganda ($n = 180$), of which 133 (26%) had microbiologically Confirmed TB, 120 (23.4%) had Unconfirmed TB (clinically diagnosed), and 231 (45.2%) were Unlikely TB cases (non-TB LRTI) based on NIH consensus definitions. To prioritize the detection of biomarkers that distinguish TB disease from other non-TB respiratory diseases, rather than non-specific inflammatory markers, our primary focus was on the comparison of Confirmed vs. Unlikely TB. We further confirmed the specificity for TB disease with a small number ($n = 27$) of asymptomatic healthy children from Uganda, of whom 8 (30%) had evidence of Latent TB infection based on a positive QuantiFERON-Gold test. Demographic and clinical characteristics are summarized in Table 1 and provided for each patient in Supplementary Data 1; the median age was 4 years (IQR 2–7), 46.4% were female, 11.2% were living with HIV, and 52.6% were underweight. Children with confirmed TB were significantly more likely to be living with HIV or be underweight than children with Unconfirmed or Unlikely TB.

### DIA-PASEF enabled high-throughput plasma proteomics

For all children, we started from 1 μL of undepleted plasma and performed high-throughput proteomics sample preparation[14] followed by data-independent acquisition (DIA-PASEF) mass spectrometry analysis (Fig. 1a)[15]. In total, we quantified 7102 peptides and 859 proteins using a high-throughput (~35 min sample-to-sample) DIA-PASEF acquisition (Fig. 1b), with an average detection of 2628 peptides and 498 proteins per sample (Fig. 1c, d and Supplementary Data 2). From this analysis, we removed 7 outlier samples showing low numbers of peptides and proteins, resulting in 504 samples in total. We achieved an average data completeness of 60.4%, with 241 proteins detected in all 504 samples and 411 detected in more than 75% of the samples (Fig. 1e). The concentration of proteins in plasma exists over a wide dynamic range exceeding 10 orders of magnitude, with a subset of proteins having very high concentrations (e.g., albumin) that can preclude the detection of lower abundance proteins. As we did not use immuno-depletion to remove proteins of high concentration[16], we evaluated the dynamic range in proteins detected in our proteomics experiments using concentration values reported from antibody and MS based-assays (HumanProteinAtlas[17]). Based on this analysis, we were able to quantify proteins spanning more than 4 orders of magnitude. While those detections were biased towards proteins of higher concentration, we were able to reproducibly detect proteins down to a level of 12.1 ng/L concentration (SERPINF2), with a median concentration of 40 ng/L (Fig. 1f).

We next evaluated our data across samples from the four clinical sites (Fig. 2a), and observed a consistent signal distribution of MS protein abundances, devoid of upper-end skewing, across 5 orders of magnitude (Fig. 2b). This resulted in highly consistent protein detections across countries, in which 88.7% of all proteins were detected across all sites, with less than 1% of all proteins displaying country-specific identification patterns (Fig. 2c). To normalize any variation between the various clinical sites, batches of sample preparation, or MS acquisition batches, we utilized COMBAT[18], a parametric approach commonly used in proteomics to mitigate batch effects[19]. We used as batches the various clinical sites, with added covariates of the MS acquisition and sample preparation batches. After normalization and COMBAT correction, we reduced our data to two dimensions using single-value decomposition to visualize the sample distribution after PCA and positively reduced batch effects for most samples as exemplified by the majority of the samples not being separated by first or second component (Fig. 2d). Lastly, from a quantitative standpoint, we achieved a low coefficient of variation (CV) both within each country (average = 7.9%) and across all countries (~8%) (Fig. 2e). Importantly, this analysis was performed using only proteins identified across more than 75% of the samples ($n = 411$) to avoid artificially decreasing the CV due to the imputation process (see Methods). Overall, this suggests the absence of substantial country-specific protein abundance differences and the possibility of using the combined data from all clinical sites for analysis of TB-specific differences.

### Identification of TB disease candidate biomarkers

We first evaluated the ability of plasma proteomics to separate healthy children from symptomatic children undergoing evaluation for pulmonary TB by comparing the protein levels of known inflammatory markers. As expected, serum amyloid protein 1, 2, 4 (SAA1, SAA2, SAA4) and C-reactive protein (CRP) were all significantly upregulated among symptomatic children, with SAA2 displaying the greatest difference amongst the acute phase proteins (Fig. 3a). However, these inflammatory markers were not able to significantly distinguish between the different groups of symptomatic children (i.e., Confirmed, Unconfirmed, or Unlikely TB) (Fig. 3a).

We next focused on comparing plasma protein levels in children with Confirmed TB ($n = 112$) and Unlikely TB ($n = 235$) to identify biomarkers that could distinguish TB disease from other non-TB respiratory diseases. From this comparison between Confirmed and Unlikely TB, we identified 47 proteins displaying significantly different abundances, of which 30 displayed downregulation and 17 displayed upregulation (Fig. 3b and Supplementary Data 3). Interestingly, one of the proteins displaying the most statistically significant regulation was the tryptophanyl t-RNA synthetase WARS1, which was increased in children with Confirmed TB vs. Unlikely TB (log2FC = 0.39, BH *adjusted p* = 3.3 × 10−5) (Fig. 3b), and is linked to TB infection via multiple mechanisms[20–22]. Overall, the majority of these are known plasma proteins with previous classifications as secreted or extracellular proteins (38/48), minimizing the possibility of random variation in tissue leakage driving the distinction between groups. For the remaining 10 (WARS1, DBH, TUBA1A, ICAM1, GSN, LTA4H, SDC1, CSF1R, THBS4, CDH13), literature evaluation of their localization demonstrated the majority being potentially secreted (9/10) with only one (TUB1A1) not having reported extracellular localization.

We further identified upregulation of multiple specific immunoglobulin heavy (IGHV1-18, IGHV1-3, IGHV2-26, IGHV3-23, IGHV3-30) and light chain variable domains (IGKV1-16, IGKV1D-33, and IGKV3-20) across several countries (Fig. 3c and Supplementary Fig. 1), potentially suggesting an oligoclonal humoral response to TB disease. Additionally, we observed significantly different levels of several proteins (APOM, PON1, CPB2) (Fig. 3b), which have been previously identified in plasma proteomic studies of severe vs non severe COVID-19[23] and an adult TB study[7], potentially pointing towards those proteins as general markers of lung inflammation rather than specific markers of pediatric TB disease.

**Table. 1 | Cohort demographic and clinical characteristics (N = 511)**

| | Age in years (median, IQR) | <5 years n (%) | 5–14 years n (%) | Female n (%) | HIV infected n (%)[a] | Underweight[c] n (%)[b] |
|---|---|---|---|---|---|---|
| Confirmed TB (n = 133, 26%) | 3 (1–7) | 77 (57.9%) | 56 (42.1%) | 64 (48.1%) | 23 (17.3%) | 83 (62.4%) |
| Unconfirmed TB (n = 120, 23.4%) | 3 (2–7) | 82 (68.3%) | 38 (31.7%) | 53 (44.2%) | 16 (13.3%) | 69 (57.5%) |
| Unlikely TB (n = 231, 45.2%) | 4 (2–8) | 129 (55.8%) | 102 (44.2%) | 109 (47.2%) | 18 (7.8%) | 117 (50.6%) |
| Healthy Control no TB infection (n = 19, 3.7%) | 5 (3.5–6.5) | 7 (36.8%) | 12 (63.2%) | 7 (36.8%) | 0 | 0 |
| Healthy Control with Latent TB infection (n = 8, 1.6%) | 4 (3–6.3) | 5 (62.5%) | 3 (37.5%) | 4 (50%) | 0 | 0 |
| All (N = 511) | 4 (2–7) | 300 (58.7%) | 211 (41.3%) | 237 (46.4%) | 57 (11.2%) | 269 (52.6%) |

[a]P-value = 0.03 for difference among Confirmed, Unconfirmed, and Unlikely TB by two-sided chi-squared testing.
[b]P-value = 0.01 for difference among Confirmed, Unconfirmed, and Unlikely TB by two-sided chi-squared testing.
[c]Underweight defined as weight-for-age Z score < −2 if less than 5 years old, or body mass index <18.5 if 5–14 years.

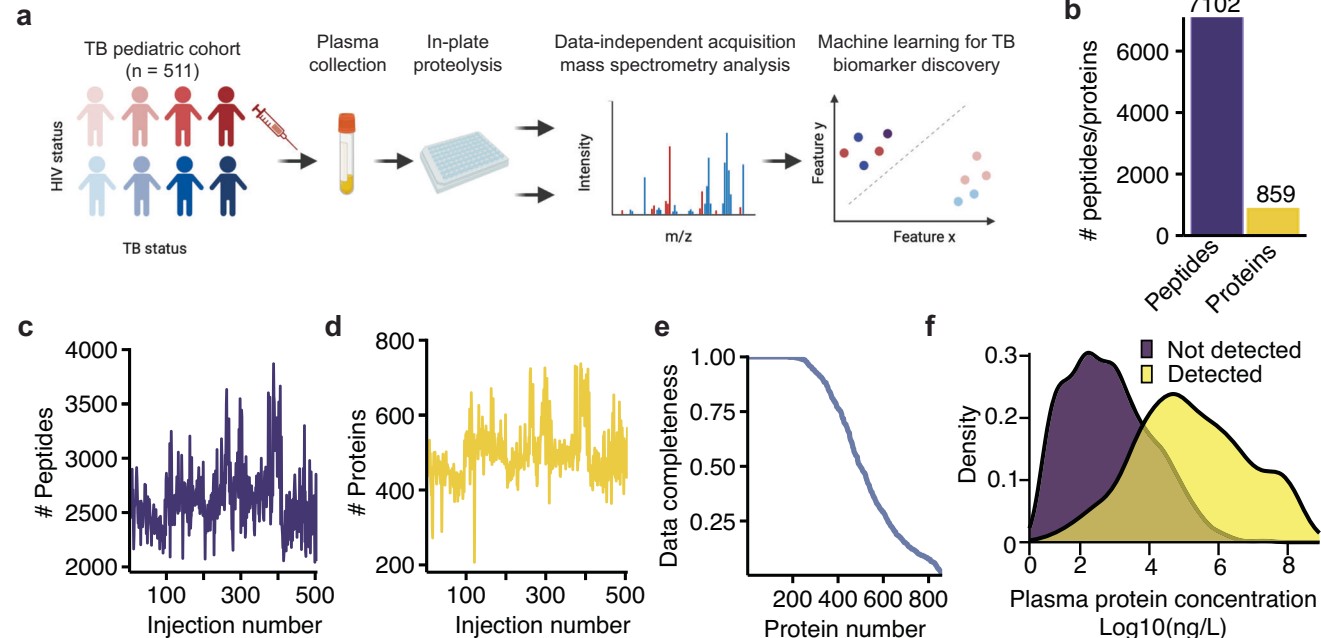

**Fig. 1 | A high-throughput workflow for plasma proteomics. a** Plasma proteomics workflow and experimental design (created by BioRender). **b** Barplot showing the total number of unique peptide sequences (purple) and protein groups (yellow) identified across all samples. **c, d** After removing 7 outlier samples, the number of peptides (**c**), and proteins (**d**) identified per MS injection. **e** Percentage of identifications (y-axis) versus the number of identified proteins (x-axis). **f** Density for the concentration ranges of plasma proteins, with those proteins detected in our study represented in the yellow density, while purple density represents remaining proteins not detected in our study. X-axis represents the logged ng/L concentration determined from HumanProteinAtlas[17]. Source data are provided as a Source Data file.

Lastly, to more broadly identify pathways with dysregulated patterns between Confirmed TB and Unlikely TB, we performed a pathway enrichment analysis on each pathway included in the KEGG and REACTOME databases, using only gene sets with more than 50% of overlap with our plasma proteomic datasets. In total, 14 pathways showed significant differential means with Benjamini–Hochberg adjusted p-value < 0.05 (Fig. 3d). Amongst the pathways showing significant regulation, we identified several related to complement activation, which have also been identified in studies of whole blood transcriptomics in TB[24,25]. Complement upregulation in the context of TB may reflect activation of the classical pathway by antigen-antibody complexes, activation of the alternative pathway or mannose-binding lectin pathway by components of the Mtb cell envelope or cell wall, and/or through increased synthesis as acute phase proteins.

### Machine learning based identification of a TB biosignature

To identify the smallest subset of features achieving the required target product profile (TPP) for a screening test (70% specificity at 90% sensitivity), we first utilized LASSO to reduce the number of features to a subset that would allow exhaustive brute-force approaches. It should be noted that prior to this analysis, proteins with more than 50% missing values between Confirmed and Unlikely TB were removed to limit the impact of the data imputation on the final biosignature. The choice of LASSO over other feature selection approaches like Tree-based or recursive feature elimination (forward or reverse) was due to the inherent sparsity of the resulting solutions and the computational performance. We utilized LASSO using 20-fold cross-validation, which led to removal of a large portion of features, resulting in 50 with non-0 LASSO coefficients (Fig. 4a and Source Data). Notably, simply selecting the top N most important proteins by their LASSO feature importance did not reach the WHO TPP for any of the N utilized (Supplementary Fig. 2), which supports the use of deep combinatorial analysis to evaluate the performance of a small subset of features.

Thus, we decided to investigate the best combination of a small subset of features using the WHO TPP as an objective function. Specifically, we calculated all possible combinations of N features (from 1

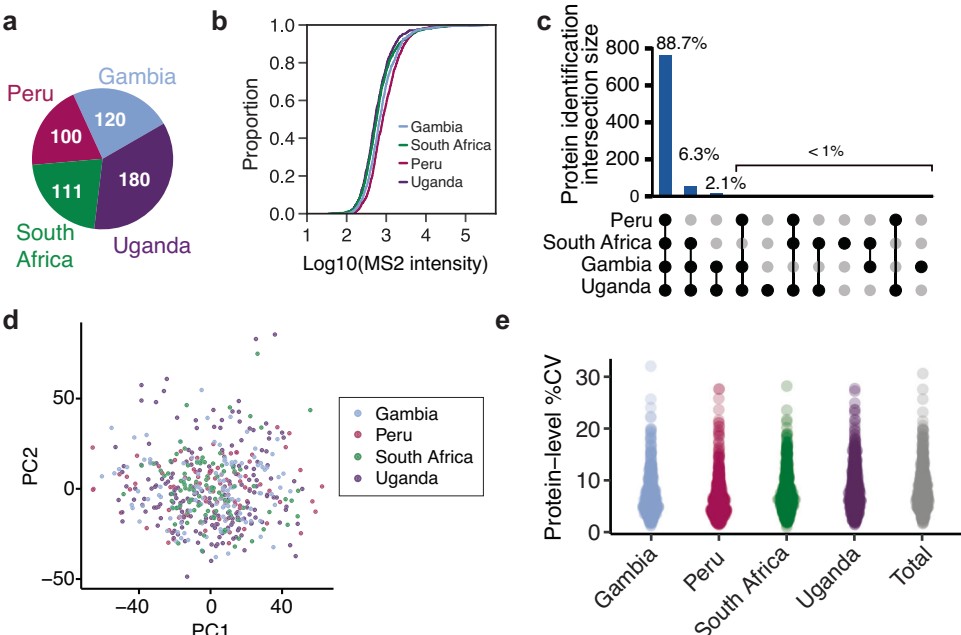

**Fig. 2 | Quality control and reproducibility of plasma proteomics across multiple clinical sites. a** Pie chart illustrating the number of samples originating from each clinical site. **b** Empirical cumulative distribution function plot for the raw MS intensity of the samples (*x*-axis) from the various clinical sites. **c** Upset plot showing the overlap in protein identifications between the different clinical sites. **d** Principal Component Analysis (PCA) of the DIA-PASEF dataset following COMBAT batch correction. *X*-axis shows the first component (10% variance) and *y*-axis the second component (6% variance). Each point represents a sample, while the color code indicates the clinical site. **e** Protein level percent coefficient of variation (%CV) within each clinical site and across all samples. Source data are provided as a Source Data file.

to 6) and selected the combinations maximizing the sensitivity at 70% specificity. For combinations achieving the same sensitivity, we selected the one with the greater AUC. We derived six logistic regression models (trained on a 75% balanced subset of the data and tested on 25% of the remaining samples), of which four met or exceeded the WHO TPP for a screening test (Fig. 4b). The 5 protein models achieved 93% sensitivity at 70% specificity (95% CI for 5 protein model 0.73–0.99), and the 6 protein models achieved 96.7% sensitivity at 70% specificity (95% CI 0.83–0.99) on our test data (Fig. 4c, *n* = 83, 30 positive, 53 negative). The derived features for the 4 to 6 protein models were mostly shared, with APOM, TNC, and CD44 being shared across the 4, 5, and 6 protein models (Fig. 4d). The selected proteins for most models showed small variance and significantly different means across all TB classes (Fig. 4e), potentially suggesting their relevance in TB disease. Two proteins further showed regulation when comparing Confirmed TB and Unlikely TB: WARS1 (log2FC 0.38, $q = 10 \times ^-5$) and APOM (log2FC −0.45, $q = 10 \times ^-5$) (Figs. 3b and 4e). Each individual protein showed a low AUC ranging from 0.577 (HEG1) to 0.745 (APOM), suggesting the lack of a single indicative feature driving the AUC and the need for at least 3 proteins to achieve the WHO TPP (Supplementary Fig. 3).

**Detection of unconfirmed TB**
We tested the derived biosignatures on 115 Unconfirmed TB cases that passed our proteomics quality control filtering to assess if we could further identify TB cases in symptomatic children with culture-negative disease. Although the lack of microbiological confirmation raises the possibility that these cases did not represent true TB disease, all children in the Unconfirmed TB group had clinical signs and symptoms of TB and improved on anti-TB treatment. In this comparison, we only used biosignatures meeting or exceeding the WHO TPP for a screening test (3, 4, 5, and 6 protein models) and utilized as a probability threshold for classification the AUC point that achieved the WHO TPP. The various models

supported the diagnosis of TB in Unconfirmed TB (negative by sputum-based testing) in ~79% of the cases, with different models predicting between 85 and 98 positive cases among the 115 children (Fig. 5a). We observed good agreement between predictions, with 73/115 samples (63%) positively predicted by all models (Fig. 5b). Importantly, we did not observe separation between healthy and latent TB when utilizing any of these three biosignatures, suggesting that these are specific for active TB disease (Supplementary Fig. 4). When evaluating the separation between the various Unconfirmed TB samples and the Confirmed TB group using all identified proteins, we observed a trend where samples positively predicted by the all four models (*n* = 70), clustered more closely to the Confirmed TB group in latent space derived by PCA, and showed separation on the first component from the negatively predicted unconfirmed samples (*n* = 11) (Fig. 5c). This suggests that we robustly extrapolated a valid biosignature as the individual contribution of these 8 proteins on the total number of proteins identified (850) is small with only TNC ranking amongst the top 20% features driving the separation on the first component (Supplementary Fig. 5).

## Discussion
This study represents the largest TB plasma proteomics study in children to date, and encompasses a diverse pediatric cohort of >500 samples across clinical sites in four LMIC and two continents. The scale of this analysis was made possible by the use of data-independent acquisition to provide high-throughput, accurate, and precise quantification of hundreds of proteins within only ~30 min of MS acquisition. This is in contrast to previous work for the development of host-based biomarker for TB using plasma proteomics, which have revolved around the use of proteomic multiplexing for quantification (e.g., ITRAQ) and long acquisition times, both of which are detrimental for acquisition of large clinical cohorts[13,26,27]. Furthermore, the power of this study is amplified by our cohort design, which includes both healthy controls and >200 controls with non-TB respiratory diseases.

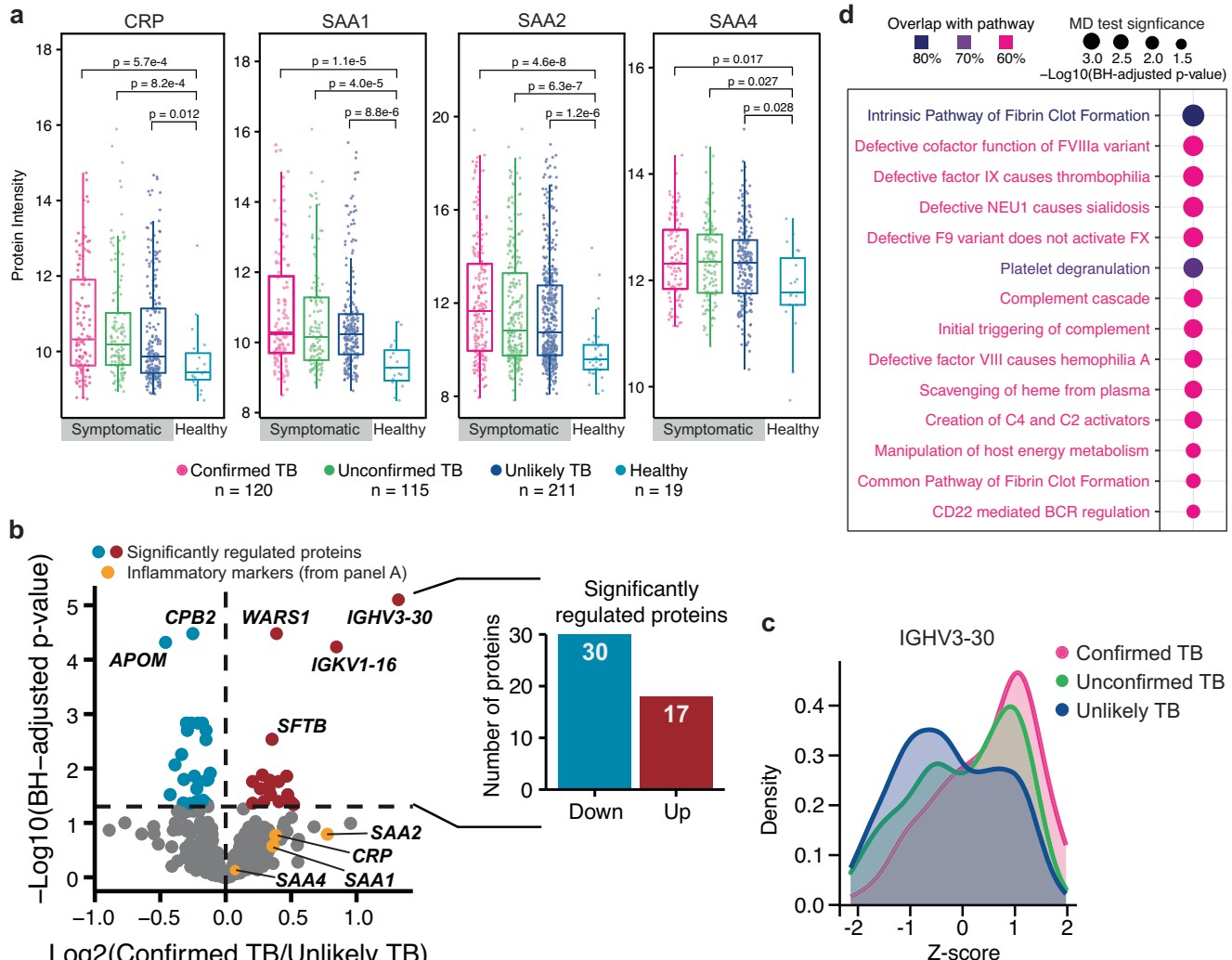

**Fig. 3 | Abundance proteomics analysis of pediatric TB cohorts. a** Benchmark of data between patients with respiratory burden and healthy controls, excluding Latent TB Infection. *X*-axis shows the TB classification status, while *y*-axis represents the protein-level intensity. Box shows the protein intensities for individual samples (dots), the median value (center line) IQR range (box limits), and 1.5 times the IQR (whiskers). *P*-values are calculated from a two-sided Kruskal–Wallis test. *N*-values represent the number of patients within each group. **b** Volcano plot between Confirmed (*n* = 133) and Unlikely TB (*n* = 231). The *x*-axis shows the Log2 fold change at the protein level, while the *y*-axis represents the significance as −log10 of the Benjamini–Hochberg (BH) corrected *p*-values derived from a two-sided Welch *t*-test. Significant proteins (BH-adjusted *p* < 5%) are shown in red (upregulated) and

blue (downregulated). Yellow dots indicate inflammatory marker proteins from (**a**). Barplot showing the number of differentially expressed proteins (DEPs) that were either upregulated (red, *n* = 17) or downregulated (blue, *n* = 30). **c** Density plot showing the *z*-scored intensity for the most significantly regulated protein (IGHV3-30), divided by TB status in confirmed TB (pink), unconfirmed TB (green), and unlikely TB (blue). **d** Gene set enrichment analysis for identification of dysregulated pathways between Confirmed TB and Unlikely TB. Dot size represents the BH adjusted *p* from a two-sided mean difference (MD) test of protein abundances. Colors indicate the overlap between each signaling pathway and the protein dataset. Only pathways with over 60% overlap are represented. Source data are provided as a Source Data file.

The inclusion of a non-TB respiratory disease control group addresses a key clinical diagnostic challenge to distinguish children with pulmonary TB disease from those with symptoms due to other causes. Inclusion of this control group avoids the detection of candidate TB biomarkers that are non-specific inflammatory markers that cannot differentiate among symptomatic states, as observed with CRP, SAA1, SAA2, SAA3, and SAA4, and which were included in previous plasma proteomic biosignatures[7,13].

An important milestone of this work is the application of machine learning to develop a minimal host-based biosignature consisting of 3–6 proteins that separate children with Confirmed TB vs. Unlikely TB at a level of specificity and sensitivity that meets or exceeds the WHO criteria for a TB screening test[10]. We found WARS1 to be a part of the 4 protein biosignature, and has been identified in adult proteomic studies as a promising TB biomarker[6,28]. WARS1 (also known as TrpRS or

SYWC) has previously been linked to TB infection by multiple mechanisms. First, upon *Mycobacterium tuberculosis* infection, a multitude of lymphocytes, including CD4 and CD8 T cells, noncanonical T cells, natural killer cells, and type 1 innate lymphoid cells upregulate[10] interferon gamma (IFNγ) as part of the host immune response, which in turn induces WARS1 expression[29]. WARS1 is also induced by tryptophan depletion[30]. Tryptophan depletion by the kynurenine pathway has been detected in multiple metabolomic studies in active TB disease[31–33], hence our data further supports previous reports on the importance of Tryptophan metabolism in active TB diseases versus other respiratory illnesses.

Several of the other proteins have either not been associated with TB or only described in adult biosignatures, which further highlights the need for pediatric-specific analyses. For example, TNC (Tenascin-C) is associated with lung disease but not specifically with TB[34]. In mice,

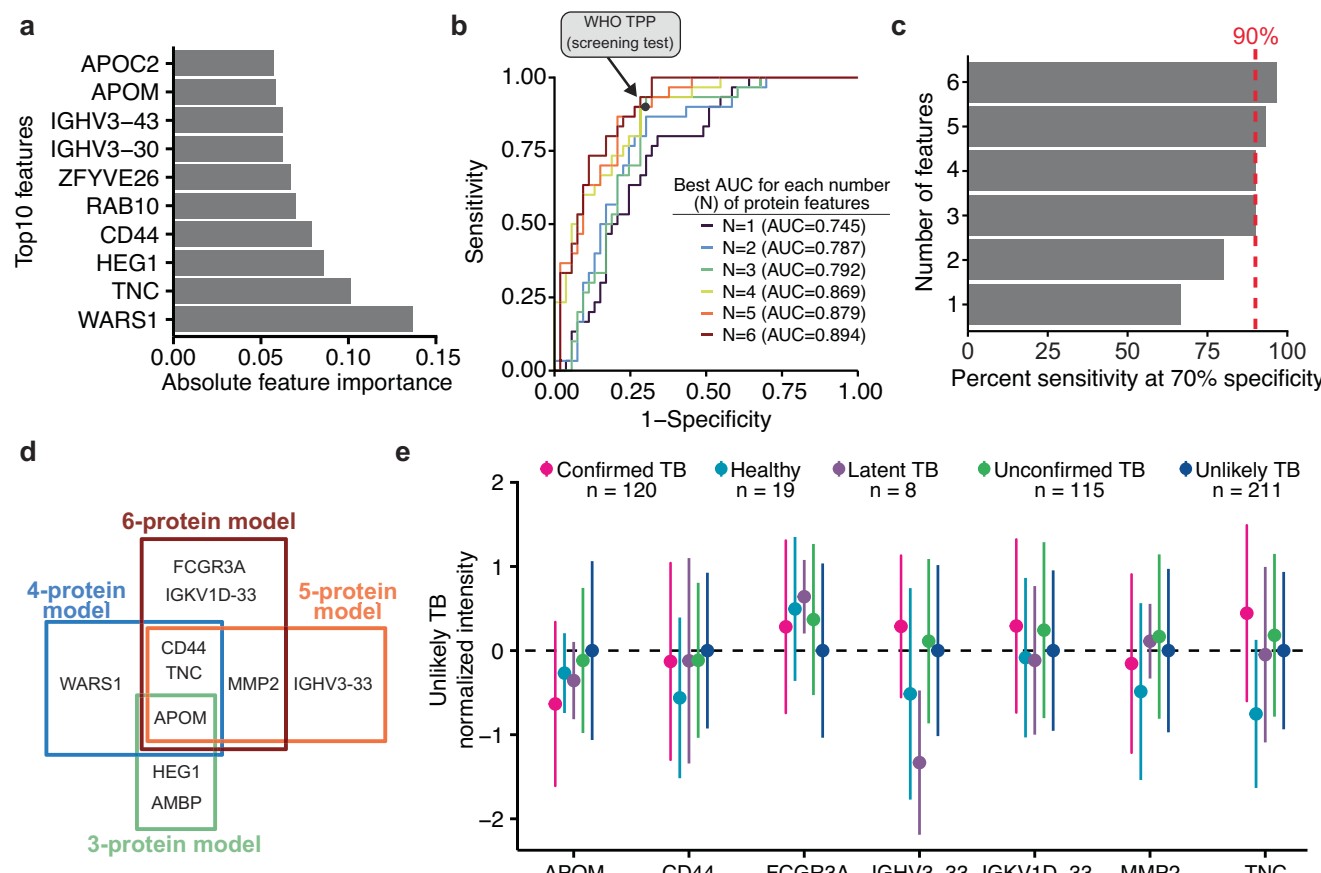

**Fig. 4 | Machine learning to develop a parsimonious biosignature for pediatric TB disease. a** Absolute feature importance from a LASSO model for the top ten most important features. **b** ROC curves for best-scoring combination of features on the test data (25%). Each curve represents the feature subset achieving the highest AUC derived from all combinations of 1 ($n = 50$), 2 ($n = 1225$), 3 ($n = 19,600$), 4 ($n = 230,300$), 5 ($n = 2,118,760$), and 6 ($n = 15,890,700$) features. WHO TPP for a screening test (70% specificity and 90% sensitivity) is denoted by the black circle. **c** Barplot for the sensitivity achieved at 70% specificity for all 6 models. Dotted red line represents 90% sensitivity. **d** Venn diagram of the overlap in proteins from the 3-, 4-, 5-, or 6-protein model. **e** Dotplot representing the mean (dot) and the standard deviation (line) for the proposed biosignature proteins (5 and 6 protein models) across individual patients from different TB classes. *N*-values represent the number of patients within each class. Different colors highlight the different TB classes according to NIH consensus definition. Each protein is normalized to the Unlikely TB protein abundance for that respective protein. Source data are provided as a Source Data file.

it has been shown that CD44 is a macrophage binding site for *M. Tuberculosis* that can provide protective immunity[35]. Further studies in adult TB patients have identified CD44 as a serum biomarker for multidrug-resistant TB in adult patients[36]. In the case of MMP-2, this protein has been studied in the context of adult TB and found to be elevated in respiratory specimens as compared to healthy controls[37,38] and correlates with markers of disease severity, such as cavitation. However, less is known about the role of this protein in biofluids such as plasma, or in pediatric patients.

APOM was also found across the signatures, and was significantly downregulated in Confirmed versus Unlikely TB. *M. tuberculosis* infection alters lipid metabolism[7,39,40], and a variety of apolipoproteins have been identified in adult proteomic studies as candidate biomarkers that are also downregulated. While APOM has not been previously reported, it is associated with HDL, which has been found to be lower in individuals with TB and correlated with radiologic extent of disease[41]. At the same time, comorbidities such as malnutrition and HIV infection increase the risk of TB and can also alter metabolism, and may have contributed to these differential markers[42]. We found that the proportion of HIV and malnutrition were higher in children with Confirmed TB, but we were limited in the sample size of our test set to perform further subgroup analyses. Prospective validation of these markers is thus needed, overall, by setting, and among key risk groups including infants, children with HIV, and malnutrition.

Importantly, our protein biosignatures did not separate between healthy children and children with latent TB infection for any of the tested models, suggesting that these protein biosignatures are specific for active TB disease. Moreover, application of these host-biosignatures to children with Unconfirmed TB was able to further support a potential diagnosis of TB in ~63% of cases that were negative by sputum-based testing. Although the lack of microbiological confirmation raises the possibility that these cases did not represent true TB disease, all children had clinical signs and symptoms of TB and improved with anti-TB treatment. However, it is important to note that we cannot know with certainty whether our biosignatures are correct in these classifications of TB among the Unconfirmed TB group. Future clinical trials in which anti-TB treatment is provided based on biosignature results would be required to fully address this question.

While the biosignatures derived for childhood TB in his study are the result of a large-scale untargeted discovery-proteomics approach, there have been several targeted cytokine-based signatures identified for TB in children[11,43,44], which are proteins that are often below the limit of detection by mass spectrometry[45]. Furthermore, in several cases, these targeted analyses were completed at a single center with a small sample size. For example, prior work identified a 3-cytokine signature to distinguish children with TB disease from other respiratory diseases in the Gambia, but they achieved a lower AUC of 0.74 and

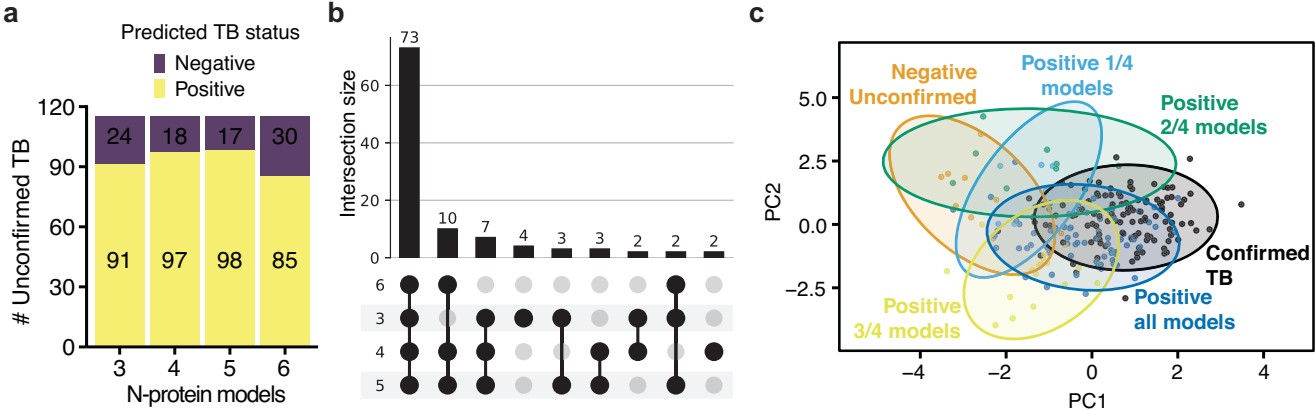

**Fig. 5 | Detection of unconfirmed TB. a** Barplot showing the number of positive predicted (yellow) and negative predicted (purple) in the proposed linear models using 3, 4, 5, or 6 proteins. Values in the barplot indicate the number of predicted cases in each category. **b** Upset plot displaying the overlap between all positive predictions using the 3-, 4-, 5-, or 6-protein models. **c** Principal component analysis of Confirmed and Unconfirmed TB. *X*-axis shows the first component and *y*-axis shows the second component. Each dot represents a sample. Samples are color coded based on either TB status (Confirmed TB, black) and further for the Unconfirmed TB based on the positive prediction in the various models: 1/4 of models (light blue), 2/4 of models (green), 3/4 of models (yellow), or all models (dark blue). Samples negatively predicted by all models (Negative Unconfirmed) are shown in orange. Shading approximates the 95% confidence region for the 2D normal distribution of each group. Source data are provided as a Source Data file.

72.2% sensitivity[44]. Our study benefited from a large sample size, representation from four countries, with a high proportion who were under five years old, living with HIV, and were undernourished. While further prospective validation and subgroup analyses are needed to evaluate robustness and reproducibility, our findings suggest that a simple host-based proteomic signature could be a valuable non-sputum TB screening test for children. To further enable translation, there is also a need for greater development of technologies to support multiplex testing at the point-of-care[46].

It is important to note that there are limitations to our study. As noted above, the power of our derived biosignatures will require further validation through, for example, prospective clinical trials in which anti-TB treatment is based on patient biosignature classifications. Our biosignatures also include immunoglobulin G proteins. While we observe highly consistent detection of these proteins across our cohort, the high degree of polymorphism in these proteins across the human population may limit their broad utility in a biosignature. Additionally, the accuracy of these biosignatures in subgroups of our cohorts was limited by sample sizes. From a technical perspective, plasma sampling, sample preparation, and data collection, may each have introduced a bias in our results. This includes our prioritization of throughput and reproducibility by not utilizing protein depletion strategies such as antibody or protein coronas. In general, we attempted to mitigate these biases by randomization across the workflow, including specimen collection and data acquisition, and post-analysis computational batch correction. Finally, our biosignatures were evaluated on the test set and not a fully independent hold-out set, hence, the reported performance may be optimistic due to multiple testing, and should be interpreted as exploratory rather than confirmatory.

In conclusion, untargeted proteomics was able to broadly evaluate the plasma of children across four countries, and identify candidate host protein biomarkers that could distinguish pediatric TB disease from other respiratory diseases. Moreover, from these candidate markers, we identified a plasma protein biosignatures of only 3–6 proteins for childhood TB disease that achieved the minimum accuracy for a TB screening tool. These efforts have provided greater characterization of the unique immune response in pediatric TB disease, while providing a non-sputum biosignature that could reduce delays in TB diagnosis and improve detection and management of TB in children worldwide.

## Methods

### Ethical considerations
This study complies with all relevant ethical regulations. All caregivers completed a written informed consent, including for storage of samples for future studies, and children completed an assent as applicable. The studies were approved by the Mulago Hospital Ethics Research Committee, Gambian Government, and MRC joint ethics committee, London School of Hygiene and Tropical Medicine, Institutional Ethics Committee for Research of National Institute of Health—Peru, University of Cape Town, and the University of California, San Francisco (UCSF) IRB.

### Pediatric TB cohort
We analyzed plasma samples that were collected from children less than 15 years old evaluated for pulmonary TB who were previously enrolled as part of prospective diagnostic cohort studies in the Gambia, Peru, South Africa, and Uganda. Children were included if they had signs and symptoms of pulmonary TB, and excluded if they were already taking treatment for TB infection or disease for more than 72 h. All children completed a standard TB evaluation, including clinical exam, chest X-ray, and respiratory sample collection for Xpert MTB/RIF molecular testing and mycobacterial culture. All children had follow-up after 2–3 months, and were assessed for clinical response to any treatment. They were classified according to NIH consensus definitions as Confirmed, Unconfirmed, or Unlikely TB. Confirmed TB was defined as having microbiological evidence of TB disease by a positive Xpert MTB/RIF Ultra or mycobacterial culture positive for *M. tuberculosis*. Unconfirmed TB cases did not have microbiological evidence of TB, but had signs and symptoms of TB disease with other clinical signs or risk factors suggestive of TB, including abnormal chest X-ray and/or known TB contact. They were started on anti-TB treatment with improvement at the follow-up visit. Unlikely TB cases were symptomatic, but did not have microbiological evidence of TB disease nor other signs or risk factors. In addition, asymptomatic healthy children from Uganda were enrolled, who had interferon-gamma release assay (IGRA) testing with Quantiferon-Gold (Qiagen, Hilden, Germany) testing for TB infection. Healthy controls were defined as asymptomatic and IGRA negative, while Latent TB infection cases were defined as asymptomatic with positive IGRA results. The gender of participants was self-reported in the baseline questionnaire, and was not considered in the study design.

## Sample collection and selection

Trained staff performed venipuncture and collected blood samples in all children at baseline and within 72 h of any TB treatment. Blood samples were centrifuged and plasma samples aliquoted and placed in −80 °C freezers. For this analysis, each study site randomly selected plasma samples from Confirmed, Unconfirmed, and Unlikely TB cases in a 1:1:2 ratio, respectively. In addition, a convenience sample of plasma specimens was selected of asymptomatic children from Uganda.

## Sample preparation for plasma proteomics

We analyzed a total of 511 plasma samples, with each sample representing an individual patient ($n = 1$). From each sample, 1 μL of undepleted plasma was transferred in a 96-well plate with 200 μL of inactivation buffer (8 M urea, 100 mM ammonium bicarbonate, 150 mM NaCl), and 0.75 μL/mL of RNAse (NEB) was added. The proteins were transferred to a 96-well filter plate and processed similarly to what we previously described[14]. Briefly, the plates were dried by centrifugation (1800 × g at 25 °C for 30 min) and 50 μL of TUA buffer (8 M urea, 20 mM ammonium bicarbonate, 5 mM TCEP) were added. Following incubation at RT on a shaker (500 rpm, 25 °C), chloroacetamide (CAA) was added to 10 mM final concentration and the plates were incubated in the dark for 1 h at room temperature. TCEP/CAA were removed by centrifugation (2000 × g, 30 min, RT) and the plates were washed thrice with 200 μL of ddH20. Trypsin was added in a 1:50 ratio and the samples were digested overnight at 37 °C on a shaker (800 rpm). Peptides were collected by centrifugation (2000 × g, 30 min at RT) and the plate was washed once with 100 μL of ddH20. Resulting peptides were dried under vacuum and were resuspended at approximately 200 ng/μL prior to MS injection and DIA-PASEF analysis. Additionally, from these samples, a representative pool of HIV positive and TB-positive cases were further high-pH fractionated on C18 tips and measured by DDA-PASEF to generate a spectral library[47]. Briefly, this high-pH fractionation was performed using C18 spin columns. These columns were first activated by treatment with one column volume of acetonitrile, followed by equilibration by two column volumes of 0.1% TFA. Peptides were subsequently loaded onto the C18 columns and washed twice with 0.1% TFA. A stepwise elution of bound peptides was performed using increasing concentrations of acetonitrile (5%, 7.5%, 10%, 12.5%, 15%, 17.5%, 20%, 50%) in 0.1% triethylamine (pH 10), and lastly with 2 washes of 50% acetonitrile. The resulting fractions were dried by vacuum centrifugation and resuspended on 0.1% formic acid prior to MS analysis by DDA-PASEF.

## DIA-PASEF data acquisition for abundance proteomics

Approx 200 ng per sample were analyzed on a Bruker TimsTOF Pro interfaced with a Ultimate 3000 UHPLC. Peptides were separated using a 15 cm PepSep column (Bruker, 150 cm length, 1.7 μm Reprosil Saphir C18 beads) and sprayed into the Captive source kept at 1700 V and 200 °C. The peptides were separated from 2 to 33% of buffer B (0.1% formic acid in acetonitrile) for 26 min, then B was increased to 90% buffer B for 5 min, and then the column was re-equilibrated at 5% buffer B for 2 min, reaching a total gradient time of 33 min. Buffer A of this separation was 0.1% formic acid. The samples were acquired in DIA-PASEF mode using nine 32 m/z DIA-PASEF windows (500–966 mz) and ion mobility between 0.85 and 1.3 Vs/cm². Data for selected samples was re-acquired when significant mass shifts were observed or when consecutive injections had reduced signal.

## DDA-PASEF and DIA-PASEF data analysis

To generate a spectral library for the analysis of DIA-PASEF data files, DDA-PASEF files were searched using MSfragger[48] within the FragPipe toolkit (v1.8) using the library generation workflow ("DIA-Speclibquant") using a human FASTA downloaded in January 2022 (20408 entries). This search was performed using tryptic cleavage specificity,

with 2 missed cleavages, fixed modification of carbamidomethylation on cysteine residues, variable modification of methionine oxidation and protein n-terminal acetylation, a precursor mass tolerance of optimized per sample ranging from −20 to +20 ppm (default in FragPipe), as product ion mass tolerance of 20 ppm, and a minimum peptide length of 8. Resulting peptide identifications were filtered to a 1% FDR at the peptide and protein level. The generated library and our previously reported plasma library[47] were merged using easypqp (https://github.com/grosenberger/easypqp). All DIA-PASEF samples were searched with DIA-NN (v1.8)[49] using a library-based strategy. MS1 and MS2 tolerances were set to 10 ppm. Protein grouping was performed based on the library ids and cross run-normalization was disabled. Following search, the global report file was filtered to <= 1% protein group Q-values ('Lib.PG.Q.Value'). Samples were excluded if the number of peptides was below 3 standard deviations of the median number of peptides (2591), which removes samples with less than 1700 peptides. The peptide-level data was normalized using median-centering of the peptides identified in all samples.

Following normalization, the missing values were imputed utilizing an heuristic strategy based on their identification frequency to leverage the large number of samples analyzed in this study.

The following rules were applied:
- Peptides identified in > 50% of the samples (at least 250 independent identifications) were imputed with the mean identification value,
- Peptides identified in <50% but > 10% of the samples were imputed utilizing a random value extracted from a generated gaussian distribution with mu and sigma of the data downshifted 1.8 × sigma
- Peptides identified in <10% of the samples were removed.

Following imputation, the peptide-level data was batch corrected using COMBAT[18] to normalize any variation between the clinical sites, batches of sample preparation, or MS acquisition batches. We used as batches the various clinical sites, with added covariates of the MS acquisition and sample preparation batches (i.e., the different plates). Peptides were rolled into proteins utilizing only proteotypic peptides and a topN strategy (max 3 proteotypic peptides per protein), using the mean intensity to represent a protein intensity. For gene set enrichment analysis, we used the MDtest function (nperm = 1000) from the GSAR R package using the protein intensity values from Confirmed and Unlikely TB samples as input[50]. Protein sets corresponding to known biological pathways were used as the input gene sets. For each signaling pathway, this function performed a two-sided mean difference test of the null hypothesis that there is no difference in the mean of a set of features (i.e., proteins) between two conditions (confirmed TB vs. unlikely TB). Resulting p-values were then adjusted by the Benjamini–Hochberg (BH) approach.

## Machine learning based identification of a TB biosignature

Protein-level intensities after normalization across all clinical sites and HIV status for Confirmed TB ($n = 120$) and Unlikely TB ($n = 211$) were selected and z-scored. For increased stringency in our proteins for biosignature development, we restricted it to only proteins with 50% or less missing values among the combined collection of patient samples from the Confirmed and Unlikely TB groups. We then selected from the remaining proteins, combinations exceeding the required WHO target product profile for a diagnostic test. Confirmed TB and Unlikely TB cases were included, given clear reference standards for TB and not TB. First, a random 75% of the data was selected for training a LASSO model using scikit-learn LASSOCv function (20 folds stratified by TB class, max_iter = 10000, tol = 0.0001). The feature importance was calculated and the proteins with non 0 coefficients were used for combinational analysis ($n = 50$ proteins). In this analysis, we generated all possible combinations of features ranging

from 1 (50 combinations) to 6 ($n = 15{,}890{,}700$ combinations) and trained a logistic regression model based on the $z$-scored abundance for each specific combination. The remaining 25% of data was then used as a test set for model evaluation for all models and was not utilized for training at any step in this initial analysis. Models for every N were ranked based on the sensitivity achieved at 90% specificity (on our 25% test split) and the top scoring models for every $N$ were kept for subsequent analysis. Confidence intervals were calculated using the Clopper-Pearson (exact binomial) method. We then applied models achieving the required WHO TPP (3, 4, 5, and 6 protein models) to the Unconfirmed TB cases to determine what proportion could be diagnosed using this model.

## Computational packages utilized

Raw proteomics data was analyzed with either MSFragger[48] (DDA data) or with DIA-NN (DIA data)[49], and the generated DDA library and our previous reported plasma library[47] were merged using easypqp (https://github.com/grosenberger/easypqp). For data processing, model training, and figure generation, we used the following packages in Python (v3.8.2): scikit-learn (v1.5.1), pandas (v2.2.2), numpy (v.1.26.4), pyCombat (v), https://github.com/epigenelabs/pyComBat, joblib (v.1.4.2), seaborn (0.13.2), matplotlib (v.3.9.2), matplotlib-base (v3.9.2), scipy (v1.13.1), statsmodel (v0.14.2). The following packages in R (v.4.3.1, release 'Beagle Scouts') were used for figure generation: ggplot2 (v.3.5.1), RcolorBrewer (v1.1.3), viridis (v0.6.5), ggpubr (v0.6.0), ggsci (v3.2.0). Additionally, the GSAR R package (v.1.40.0) was used for analysis of the log2FC between Confirmed and Unlikely TB. All code for data analysis, imputation, and figure plots is available here: https://github.com/anfoss/COMBO_code.git.

## Statistics and reproducibility

We randomly selected plasma samples in a 1:1:2 ratio of Confirmed:Unconfirmed:Unlikely TB, and sample size was determined by availability of specimens and to ensure adequate precision in the test set. With a sample size of 500 and 25% held for the test set, we would be powered to measure a sensitivity of 90% +/− 12% and specificity of 70% +/− 10% when comparing Confirmed to Unlikely TB. Samples were batched by country, and randomized within a given sample preparation plate and data acquisition for each country and staff were blinded to TB status during data acquisition. All samples were analyzed once with the exception of selected samples where there was evidence of instrument performance deviation, including the observation of significant mass shifts or consecutive injections with reduced signal. Data for these samples was re-collected, and this re-collected data is presented in this study. Samples not passing QCs defined in the section "DIA-PASEF enabled high-throughput plasma proteomics" were removed ($n = 7$). In the machine learning analysis, data were excluded for greater than 50% missingness.

## Reporting summary

Further information on research design is available in the Nature Portfolio Reporting Summary linked to this article.

## Data availability

The raw and processed MS data generated in this study has been deposited in the MassIVE repository with the following dataset Identifier: MSV000096394 and in the ProteomeXchange with the following dataset identifier: PXD057814 with the https://doi.org/10.25345/C5F18SS6N. Source data are provided with this paper.

## Code availability

All code for data analysis, imputation, and figure plots is available here: https://github.com/anfoss/COMBO_code.git and at https://doi.org/10.5281/zenodo.15591003.

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

## Acknowledgements

The authors are grateful to children and parents for their participation, and to the clinicians and staff at our multiple clinical sites who provided care for the children in this study. NIH funding R01AI152161 and R01AI175312 to A.C., J.D.E., and D.L.S., NIH K23HL153581 to D.J., NIH K23AI144040 to J.M.C., and NIH U19AI109755 to M.F.F. and RCalderon. UKRI MR/P024270/1 and MR/K011944/1 funding to B.K. South African Medical Research Council (SA-MRC) funding to H.Z. We also extend our gratitude to the Paul Farmer African Initiative for Research (PFAIR) for enabling a collaborative partnership between researchers from institutions in Africa and North America to exchange diverse perspectives that enrich research and drive discovery. Figure 1A was created in BioRender by AF (2025) https://BioRender.com/o99n413.

## Author contributions

Conceptualization: D.L.S., D.J., A.C., J.D.E. Data curation: A.F., J.M., P.W., R.Castro, J.L., E.N., M.F.F., B.K., E.W., A.C., D.J., H.Z. Formal analysis: A.F., J.M., R.Castro, R.N., Z.M. Funding acquisition: D.L.S., J.D.E., A.C., D.J., J.M.C. Investigation: A.F., P.W., R.Calderon, J.L., E.N., M.F.F., B.K., E.W., A.C., J.D.E., D.J., H.Z., A.M. Methodology: A.F., D.J., D.L.S., J.D.E., A.C., J.C., G.B.S., M.R.S. Project administration: D.L.S., A.C., J.D.E., B.K., E.W., H.Z. Software: A.F. Resources: J.D.E., A.C., D.J., H.Z., M.F.F., R.Calderon, B.K., E.W. Supervision: D.L.S., J.D.E., D.J., A.C., M.R.S., B.K., E.W., H.Z. Validation: A.F., J.M. Visualization: A.F., J.M., D.L.S., R.N., Z.M. Writing–original draft: A.F., D.L.S., J.D.E., A.C., D.J., R.N., Z.M. Writing–review & editing: All authors.

## Competing interests

The authors declare no competing interests.

## Additional information

[1]J. David Gladstone Institutes, San Francisco, CA, USA. [2]Quantitative Biosciences Institute (QBI), University of California San Francisco, San Francisco, CA, USA. [3]Department of Cellular and Molecular Pharmacology, University of California San Francisco, San Francisco, CA, USA. [4]Uganda Tuberculosis Implementation Research Consortium, Walimu, Kololo, Kampala, Uganda. [5]Institute for Global Health Sciences, Center for Tuberculosis, University of California San Francisco, San Francisco, CA, USA. [6]Department of Pediatrics, Division of Pediatric Infectious Diseases, University of California San Francisco, San Francisco, CA, USA. [7]Advanced Research and Health, Lima, Peru. [8]Department of Medicine, Division of Pulmonary and Critical Care Medicine, University of California San Francisco, San Francisco, CA, USA. [9]Department of Medicine, Division of Experimental Medicine, University of California San Francisco, San Francisco, California, CA, USA. [10]Department of Pediatrics and Child Health, South African Medical Research Council Unit on Child and Adolescent Health, University of Cape Town, Cape Town, South Africa. [11]Department of Pediatrics, Dora Nginza Hospital, Gqeberha, South Africa. [12]Vaccines and Immunity Theme, MRC Unit The Gambia at the London School of Hygiene and Tropical Medicine, Fajara, The Gambia. [13]Department of Global Health and Social Medicine, Harvard Medical School, Boston, MA, USA. [14]Division of Infectious Diseases, Department of Medicine Solna, Karolinska Institutet, Stockholm, Sweden. [15]Division of Infectious Diseases, Department of Medicine, Emory University School of Medicine, Atlanta, GA, USA. [16]Meso Scale Diagnostics, LLC., Rockville, MD, USA. [17]Department of Epidemiology and Biostatistics, University of California San Francisco, San Francisco, CA, USA. [18]Charité Center for Global Health, Charité Universitätsmedizin Berlin, Berlin, Germany. [19]Division of Pulmonary Diseases and Critical Care Medicine, Department of Medicine, University of California Irvine, Irvine, CA, USA. [20]Present address: Department of Global Health, Rollins School of Public Health, Emory University, Atlanta, GA, USA. [21]These authors contributed equally: Andrea Fossati, Peter Wambi, Devan Jaganath. ✉e-mail: heather.zar@uct.ac.za; danielle.swaney@ucsf.edu

## the COMBO Study Consortium

**Andrea Fossati**[1,2,3,21], **Peter Wambi**[4,21], **Devan Jaganath**[5,6,21], **Roger Calderon** [7], **Robert Castro**[5,8], **Alexander Mohapatra** [5,9], **Justin McKetney** [1,2,3], **Juaneta Luiz**[10,11], **Rutuja Nerurkar**[5,6], **Esin Nkereuwem** [12], **Molly F. Franke** [13], **Zaynab Mousavian**[14,15], **Jeffrey M. Collins**[16], **George B. Sigal** [17], **Mark R. Segal**[18], **Beate Kampman** [12,19], **Eric Wobudeya**[4], **Adithya Cattamanchi**[5,20], **Joel D. Ernst** [5,9], **Heather J. Zar**[10] ✉ & **Danielle L. Swaney** [1,2,3] ✉

A full list of members and their affiliations appears in the Supplementary Information.

