## [Transparent Peer Review file · Nature Communications]

Plasma proteomics for biomarker discovery in childhood tuberculosis

Corresponding Author: Dr Danielle Swaney

Version 0:

Reviewer comments:

Reviewer #1

(Remarks to the Author)

The aim of the study described in the manuscript "Plasma proteomics for novel biomarker discovery in childhood tuberculosis" is relevant and with impact in the diagnosis of childhood tuberculosis. Unfortunately, although several important researches were conducted with the proposal of discovering biomarkers for the diagnosis of TB in adults, those were not further explored to create a product with point-of-care use.

This study includes several positive aspects, like the size of the cohort, the experimental design, the use of a high-throughput experimental workflow (described with enough detail), the extensive and systematic analysis of the experimental data. This combination makes brings to the results from this study relevant novel information. The discussion and conclusion are supported by the experimental results.

No major faults were found in the manuscript. Below a few minor suggestion to improve the work are provided.

The number of healthy controls is very small (3,7 and 1,6%, making a total of 5,3%) compared with the TB groups. This fact deserves a comment.

Page 3: "From this analysis, we removed 7 outlier samples showing low numbers of peptides and proteins, ..." The established acceptance criteria should be provided.

Page 16: The composition of buffer A used for LC separation of the tryptic peptides should be provided.

Reference 23 is missing from the reference list

The names of the supplementary files should be made more comprehensive.

The Massive Repository username and password provided did not granted the access to the data.

Reviewer #2

(Remarks to the Author)

Paediatric TB has a major global burden and presents a major diagnostic challenge. Therefore, this is a very important research area.

A major strength of the study is the number of children enrolled (511), the global distribution of cases and the comparison against non-TB respiratory disease. The non-depletion methodology is also a benefit. Specific comments are below.

1. A limitation is the lack of a validation cohort. However, I feel that the size and quality of the study cohort, and the challenges of studying paediatric TB, compensates for this. Is there a complementary bioinformatic analysis approach that could be used to perform parallel analysis that then reaches similar biomarkers, thereby increasing confidence in the findings? This would significantly improve confidence that these are the lead biomarkers.
2. Did the children living with HIV separate from those not on the PCA? This seems important to determine whether different biomarkers may be needed depending on immune status.
3. I found it slightly counterintuitive that Fig 3B came before C and D. Can the authors clarify why the pathway analysis preceded the analysis of single genes, which would seem more logical? As a minimum, greater explanation of the approach

is made as how the pathway analysis was performed is not clear.

4. Fig 5: The implication seems to be that some unconfirmed cases can be identified as TB, but as there is no firm case diagnosis then these findings need to be interpreted very carefully. As a minimum, can more detail be given about these cases (i.e. were they all treated for TB, what were their outcomes)? I am not convinced that this figure further supports the validity of the model without more information about the cases, purely by the nature of the cases being “unconfirmed”.

5. Discussion: it is important to note that there is no near-patient device that can measure multiple markers highly accurately, and so this represents a major technical challenge to the implementation of any new plasma based TB diagnostic.

Reviewer #3

(Remarks to the Author)

Summary:

Diagnosis of pediatric TB remains incredibly challenging, especially in very young children from whom respiratory samples are hard to obtain and who typically present with paucibacillary disease. Biomarkers for diagnosis in this population are lacking, and especially in young children, could be unique from adult populations, in whom the majority of work has been done to date. A major strength of this study is that it moves the discovery effort to the relevant population, children living in TB endemic populations. Specifically, to discover protein signatures that are suggestive of TB disease, mass spectrometry was performed on over 500 blood samples from children with confirmed TB disease, unconfirmed TB and non-TB respiratory disease. Another major strength of the study is that the major comparison is between confirmed TB disease and non-TB respiratory disease, the clinical scenario in which a pediatric-specific biomarker would be most useful. Regarding this comparison, the authors find signatures of 4-6 proteins that seemingly predict pediatric TB with reasonable accuracy (AUC 0.86-0.88). If validated and translated, this would represent a significant advancement in the field. However, as currently written, the significance, validity of data, methodology, and analytical approach are difficult to evaluate for the reasons detailed below.

Major strengths:

The study design supports the potential significance of the results. Specifically, the study is performed in children, the majority of whom are young children, living in multiple countries in which TB is endemic. Given that mass spectrometry was performed for protein analysis, the sample size is impressive. The potential significance is further supported by the clinically relevant non-TB respiratory disease control arm. AUCs of the proteomic signatures are impressive (though clarity is needed as to if these are the AUCs for the validation cohort only, or for all samples, as discussed below). Thus, especially as there are far fewer proteomic, as compared to transcriptomic, biomarkers studies in general and even fewer proteomic biomarker studies include young children, this study has the potential to be highly impactful to the field.

Critiques and suggested improvements:

Major critiques

- The argument for this study would benefit from being more flushed out and better referenced. Although pediatric biomarkers are a major gap in the field, this is not entirely clear from what is written. The introduction would benefit from a deeper literature review addressing the lack of discovery studies done in children and paucity of adult biomarkers that have been studied and shown to be valid in children.
- The study would benefit from sub-group analyses, including age <2, sex, HIV status, and nutritional status. For example, there are major differences in HIV status and nutritional status between TB and unlikely TB, and while this is not unexpected, it should be acknowledged and discussed. Furthermore, subgroup analysis may help to determine if these are playing a role in your signature. Finally, Table 1 does not include any statistical analysis of potential differences between cohorts.
- Did you use a program to analyze your data? If so, where is the code? You refer to a github page about a python package, but there is not code for analysis in python. Did you use any other programs for analysis or graphing?
- Statistics used are lacking in some places. There is no methods section on statistics and some components are missing as to what test was used. For example, what statistical test did you use for your volcano plot in Figure 3C?
- The validity of the study is somewhat hard to judge currently, because it is unclear exactly what you have done. The main point of confusion for us was whether or not the model was trained on 75% of the data and then independently validated on 25% of the data? And if so, is the figure 4B related to this and the AUC generated only on the 25% validation cohort? If so, this should be made clear. If not, then we are concerned about the need for validation of the signatures found.
- Figure 5 is difficult to interpret. If we assume that unconfirmed TB is mainly truly TB, then the signature did poorly. If, however, we assume the more likely scenario that unconfirmed TB is a mixed picture, then how are we supposed to know your signature is correctly determining the children with TB? Since the children are followed prospectively, is there more information about their long-term outcomes? Is there a different way to validate your data? In addition, regarding Figure 5C, it is not at all clear that there is closer clustering as suggested. Rather, positive in 2/3 of the models appears to be closer than all models. Furthermore, there is significant deviation in the PC2 from all models, that ends up being a similar degree with PC1 for negative unconfirmed. Therefore, there is overstatement of your conclusions here.
- The discussion should include comparison with the results of other biomarker studies. If other biomarker studies, particularly other pediatric studies, show similar proteins to your study, this discussion should be added. For example, comparison of the results of this study to the study of a three-marker protein biosignature that distinguishes tuberculosis from other respiratory diseases in Gambian children should be discussed (Togan et al, EBioMedicine, 2020 Aug;58:102909). We cannot see if any of the genes from this study are differentially expressed in your study. If so, how did this model do in your cohorts? Does it predict similar children to have TB in the unconfirmed TB group? If these proteins are not found in your data, why might it be different?
- The references need significant work. For example, reference 23 cited in the discussion, is not included in the references

section. Reference 22 in the methods does not appear to be the same reference as reference 22 in the discussion. Reference 19 in the discussion refers to a WHO reference but would not be an appropriate reference for IFN γ production by immune cells during Mtb infection yet is also listed here. Reference 18 is out of order, appearing later than reference 19-23 in the text. There are also missing references including statistics that are unreferenced but should be - e.g. "96% of deaths in children are in those whom treatment is not yet initiated" and "...estimated half of the children with TB diseases....".

Minor critiques

- Regarding enrollment into the cohorts, a workflow of how many children were evaluated and included/excluded would add clarity. Also, specifics of why children were excluded should be added.
- Regarding Supplemental Table 1, it seems there is missing data for some of the donors from The Gambia: C337, C371 and C389-C400. All healthy and latent TB donors are from Uganda. This should be mentioned in the text.
- Some components that should only be in the methods appear in the results and are somewhat distracting. As examples, the quantity of plasma, the filter-based processing in 96 well plates, and the type of mass spectrometry machine used are appropriately included only in the methods section.
- Regarding Figure 1, does the data shown in Figure 1B represent the total number of peptides and proteins, or is this the average for each individual? For the 7 outlier samples that were removed, how different were the peptide and protein numbers in these samples? Is there a standard number that would usually be used as a cutoff for this kind of analysis? Are these 7 included in figure C/D? Figure 1F is difficult to follow. Have you looked for proteins that are greater than or less than 4 orders of magnitude from the reference levels? And if so, what is the range of reference levels? What would be the expected distribution of these proteins in an otherwise healthy population? Finally, "over >4 orders" is redundant. Should be just over 4 or > 4. What is SERPINF2? Is this the lowest abundance protein?
- Regarding Figure 3C, what is meant by manual curation of their localization?
- Regarding Supplementary figure 2, what is TopN vs BestN?
- You may consider adding to the discussion that the lack of differences between countries may actually validate doing single-country studies.
- The discussion would benefit from the addition of a discussion of limitations.
- Regarding references, #10-12 appear in the text before #9. "WARS1 . . . linked to TB infection via multiple mechanisms" should be referenced.

Reviewer expertise:

- Senior reviewer – expertise in development of pediatric TB biomarkers and in pediatric infectious diseases including tuberculosis.
- Junior reviewer – expertise in TB and in analytic methods.

Reviewer #4

(Remarks to the Author)

Reviewer #5

(Remarks to the Author)

The paper reports interesting, original and important results in the field of paediatric tuberculosis, and features a good sample size and rational approach to discover a protein signature for sputum-free diagnosis. The inclusion of 4 countries and symptomatic controls is a great strength, as pointed out by the authors.

The paper is well-written and communicates its message concisely.

My main concerns are with the reporting of the data analysis. Below, please see requests for clarification and more detailed reporting of the analysis methods. I recommend the authors use their own discretion whether to add the requested details to the main or supplementary text.

The imputation strategy is clearly explained, but no motivation for it is provided. I am not experienced in proteomics, so it may be that this strategy is standard in the field. It is not clear to me why different imputation strategies were used for values missing in more, or less than 50% of samples. It may also not be clear to other readers. Please provide the rationale for these strategies.

Please provide a clearer explanation of how the 75% and 25% split data was used. It is clear that the 75% was used for the feature selection with LASSO. But after that, for the all-combinations models, how were the final 6 models selected? Was every one of them applied to the 25% dataset and the final 6 were the best ones? Or were they somehow ranked by performance on the 75% with cross-validation and then only the best 6 were applied to the 25%?

If every single model of the thousands of combinations was applied to the 25% dataset, this has to be made very clear in the Methods and discussed as a limitation in the discussion. The function of a data split is to provide an "untouched" dataset to evaluate overfitting of trained models. If the untouched dataset is utilized for every single trained model, it essentially becomes another training step and not an independent evaluation of the model. The results should be treated with more caution in this case, as an independent evaluation step to confirm the performance, is still lacking.

Please make clear for all the results, whether it is all data, the 75% or the 25% result that is reported in every case.

How was the training performed? Please provide specifics on the type of cross-validation and the software used. It is only stated that scikit-learn was used for the LASSO step.

For the combinatorial analysis, it is stated that linear models were used with different combinations of proteins. Linear regression models do not output a classification score, so it is not clear how linear models could have been used in this

step? It is possible that generalized linear models were used with the binomial family and logit link? Please provide details of the model that was used, the software and function, and provide a generic model formula to clarify. Finally, please report the original percentage missingness for each of the markers included in the final models, as this will also influence the confidence in the performance.

Version 1:

Reviewer comments:

Reviewer #2

(Remarks to the Author)

I have read the responses to my comments, which I feel have all been addressed, and also looked at the other reviewer comments and the revised manuscript. The issues raised by the reviewers have been addressed satisfactorily in my opinion, apart from final copyediting during the production process.

Reviewer #3

(Remarks to the Author)

We appreciate the detailed response to our critiques and have no further concerns.

Reviewer #4

(Remarks to the Author)

Reviewer #5

(Remarks to the Author)

Thank you to the authors for the detailed feedback on the previous review comments. Most comments were addressed and the analysis methods have been made clearer in the paper. The authors are also commended for now providing the analysis code in a GitHub repo.

One outstanding issue is the fact that every single model that was trained, was also tested on the test set. It is more robust to train models, then apply them through cross-validation, still only on the training set (or a hold-out set, separate from the test set). Then rank them according to their cross-validation performance on the training set, then select the top 3, or 5 which are then the only ones applied to the test set. In your case, you could have selected the top 2 cross-validated models from every N markers and only applied those to the test set.

The test set serves as protection against overfitting and multiple testing. By applying every single model to the test set, it has basically now become part of the training, and your results are essentially 4-fold cross-validation results. Because of random variation, you would expect 5% false positives, and from millions of potential models, that becomes a huge number. I request the authors to acknowledge this in the limitations section - because every trained model was applied to the test set, the final model results are essentially cross-validation results and have not been conclusively validated on an independent cohort. This does not take away anything from the many strengths of this study, most notably the good sample size and geographic diversity.

The authors may also consider discussing why only logistic regression was attempted for these classification models. Tree-based methods and non-linear methods could also have been evaluated and may have achieved better AUCs.

Another useful addition could be to show correlations between all the markers included in all the final models. It may explain why some markers appear in only one model and could aid in taking these markers forward into a different technology. The authors may use their discretion to include this.

Reviewer #1 (Remarks to the Author)

The aim of the study described in the manuscript “Plasma proteomics for novel biomarker discovery in childhood tuberculosis” is relevant and with impact in the diagnosis of childhood tuberculosis. Unfortunately, although several important researches were conducted with the proposal of discovering biomarkers for the diagnosis of TB in adults, those were not further explored to create a product with point-of-care use.

This study includes several positive aspects, like the size of the cohort, the experimental design, the use of a high-throughput experimental workflow (described with enough detail), the extensive and systematic analysis of the experimental data. This combination makes brings to the results from this study relevant novel information. The discussion and conclusion are supported by the experimental results.

No major faults were found in the manuscript. Below a few minor suggestions to improve the work are provided.

We thank the reviewer for acknowledging the novelty and rigor of the work presented in our manuscript.

The number of healthy controls is very small (3,7 and 1,6%, making a total of 5,3%) compared with the TB groups. This fact deserves a comment.

We have modified a statement in the “Clinical Characteristics of the Cohort” to emphasize the utility of the healthy controls in this cohort:

“ To prioritize the detection of biomarkers that distinguish TB disease from other non-TB respiratory diseases, rather than non-specific inflammatory markers, our primary focus was on the comparison of Confirmed vs. Unlikely TB. We further confirmed the specificity for TB disease with a small number (n = 27) of asymptomatic healthy children from Uganda, of whom 8 (30%) had evidence of Latent TB infection based on a positive QuantiFERON-Gold test. ”

Page 3: “From this analysis, we removed 7 outlier samples showing low numbers of peptides and proteins, ...”
The established acceptance criteria should be provided.

We provide more details on the proteomics sample preparation to ensure clear understanding of our acquisition criterion and procedure. We re-acquired plates where we observed significant mass shift during analysis or where we observed consecutive injections with low signal intensities (which is most often attributed to HPLC-MS performance). This was done to ensure that we did not remove samples based on instrument performance alone. Following re-acquisition most plates resulted in a similar number of peptides and proteins, besides seven which showed <1000 peptides. Before analysis we set on using 3 standard deviations as cut-off as this will retain ~99% of the data while removing clear outliers.

To clarify the exclusion criteria for outlier samples, we added the following statement to the “DIA-PASEF data analysis” section:

“Samples were excluded if the number of peptides was below 3 standard deviations of the median number of peptides (2591), which removes samples with less than 1700 peptides”.

We have also noted in the “Data acquisition for abundance proteomics section” that:

“Data for selected samples was re-acquired when significant mass shifts were observed or when consecutive injections had reduced signal.”

Page 16: The composition of buffer A used for LC separation of the tryptic peptides should be provided.

A sentence stating that “Buffer A of this separation was 0.1% formic acid.” has been included in the methods section.

Reference 23 is missing from the reference list

We apologize for the oversight and have corrected our reference list.

The names of the supplementary files should be made more comprehensive.

We added descriptions to the supplementary files for each column

The Massive Repository username and password provided did not grant the access to the data.

We apologize for this mistake. There was a typo in the username we provided.

The correct credentials have been updated in the manuscript and are listed below:

Dataset Identifier: MSV000096394

Reviewer Username: MSV000096394_reviewer

Reviewer Password: H6DVzA8T79W1RC6c

Reviewer #2 (Remarks to the Author)

Paediatric TB has a major global burden and presents a major diagnostic challenge. Therefore, this is a very important research area.

A major strength of the study is the number of children enrolled (511), the global distribution of cases and the comparison against non-TB respiratory disease. The non-depletion methodology is also a benefit. Specific comments are below.

We appreciate this positive feedback and agree that our study is uniquely powerful in its large cohort size that spans a globally diverse population.

1. A limitation is the lack of a validation cohort. However, I feel that the size and quality of the study cohort, and the challenges of studying paediatric TB, compensates for this. Is there a complementary bioinformatic analysis approach that could be used to perform parallel analysis that then reaches similar biomarkers, thereby increasing confidence in the findings? This would significantly improve confidence that these are the lead biomarkers.

We agree that a validation cohort would be beneficial. This work is planned, but is still several years away from completion, and thus is beyond the scope of this initial study. As such, we have noted this as a future direction in the discussion section. With regard to bioinformatic approaches to increase the confidence of the findings, we have used a common approach of splitting our dataset into a “training” and a “test” set for biomarker signature development and assessment. Here, we randomly selected 75% of our data to train the model. Then the model was tested on the remaining 25% of the data, which was not used in the training. This 25% of the data serves the bioinformatic function of a “validation cohort”, as the model is naive to this specific data. The accuracy of the model was evaluated using this 25% test set, before further applying the model to the entire dataset to retrieve specific biomarker protein sets. This approach is now more fully detailed in the “Machine learning for identification of a proteomic biosignature for childhood TB disease” section of the manuscript.

2. Did the children living with HIV separate from those not on the PCA? This seems important to determine whether different biomarkers may be needed depending on immune status.

We did not observe clear separation depending on the HIV status (see attached figure), albeit we recognize the significantly higher number of HIV negatives compared to HIV positive in this subgroup analysis. This might

affect the PCA representation due to a large statistical variation expected to increase linearly with the number of children.

3. I found it slightly counterintuitive that Fig 3B came before C and D. Can the authors clarify why the pathway analysis preceded the analysis of single genes, which would seem more logical? As a minimum, greater explanation of the approach is made as how the pathway analysis was performed is not clear.

We appreciate this feedback and have now changed the order of figure panels and their associated text. What was originally Fig. 3B is now Fig. 3D. We have also added the following text to the methods section to provide more details on how the pathway analysis was performed:

“For gene set enrichment analysis, we used the MDtest function (nperm=1000) from the GSAR R package using the quantification between Confirmed and Unlikely TB as the input values. For each signaling pathway, this function performed a mean difference test of the null hypothesis that there is no difference in the mean of a set of features (i.e., proteins) between two conditions (confirmed TB vs. unlikely TB). Resulting p-values were then adjusted by the Benjamini-Hochberg approach.”

4. Fig 5: The implication seems to be that some unconfirmed cases can be identified as TB, but as there is no firm case diagnosis then these findings need to be interpreted very carefully. As a minimum, can more detail be given about these cases (i.e were they all treated for TB, what were their outcomes)? I am not convinced that this figure further supports the validity of the model without more information about the cases, purely by the nature of the cases being “unconfirmed”.

All children with Unconfirmed TB were treated for TB, and at follow up 2-3 months later they had improved. The reviewer is correct that (1) we assume that the unconfirmed cases represent a mixed population of both children with and without TB, and (2) we cannot know with certainty whether our signature is correctly determining the children with TB among the Unconfirmed TB group, as doing so would require a trial in which anti-TB treatment is provided based on signature results. Alternatively, future studies could consider further phenotyping (e.g., with transcriptomics) to see if these results also support a diagnosis of TB. For now, all we can say is that we were able to identify a “TB signature” in many, but not all children in the Unconfirmed TB group, likely supporting that the Unconfirmed TB group is a mixed picture as noted by the reviewer. We have included text to this effect in the manuscript, particularly to emphasize this uncertainty.

We have expanded upon this by adding the following text to the discussion:

“Moreover, application of these host-biosignatures to children with Unconfirmed TB was able to further support a potential diagnosis of TB in ~63% of cases that were negative by sputum-based testing. Although the lack of microbiological confirmation raises the possibility that these cases did not represent true TB disease, all children had clinical signs and symptoms of TB and improved with anti-TB treatment. However, it is important to note that we cannot know with certainty whether our biosignatures are correct in these classifications of TB among the Unconfirmed TB group. Future clinical trials in which anti-TB treatment is provided based on biosignature results would be required to fully address this question.”

Additionally, we have included additional text in the Methods section stating that all children in the Unconfirmed group were started on anti-TB treatment with improvement at the follow-up visit.

5. Discussion: it is important to note that there is no near-patient device that can measure multiple markers highly accurately, and so this represents a major technical challenge to the implementation of any new plasma based TB diagnostic.

Thank you for this. We have added to the discussion that our results support the need for further development of multiplex point-of-care assays in parallel to ensure rapid translation, specifically this text reads as follows: *“our findings suggest that a simple host-based proteomic signature could be a valuable non-sputum TB screening test for children. To further enable translation, there is also a need for greater development of technologies to support multiplex testing at the point-of-care.”*

Reviewer #3 (Remarks to the Author):

Summary:

Diagnosis of pediatric TB remains incredibly challenging, especially in very young children from whom respiratory samples are hard to obtain and who typically present with paucibacillary disease. Biomarkers for diagnosis in this population are lacking, and especially in young children, could be unique from adult populations, in whom the majority of work has been done to date. A major strength of this study is that it moves the discovery effort to the relevant population, children living in TB endemic populations. Specifically, to discover protein signatures that are suggestive of TB disease, mass spectrometry was performed on over 500 blood samples from children with confirmed TB disease, unconfirmed TB and non-TB respiratory disease. Another major strength of the study is that the major comparison is between confirmed TB disease and non-TB respiratory disease, the clinical scenario in which a pediatric-specific biomarker would be most useful. Regarding this comparison, the authors find signatures of 4-6 proteins that seemingly predict pediatric TB with reasonable accuracy (AUC 0.86-0.88). If validated and translated, this would represent a significant advancement in the field. However, as currently written, the significance, validity of data, methodology, and analytical approach are difficult to evaluate for the reasons detailed below.

Major strengths:

The study design supports the potential significance of the results. Specifically, the study is performed in children, the majority of whom are young children, living in multiple countries in which TB is endemic. Given that mass spectrometry was performed for protein analysis, the sample size is impressive. The potential significance is further supported by the clinically relevant non-TB respiratory disease control arm. AUCs of the proteomic signatures are impressive (though clarity is needed as to if these are the AUCs for the validation cohort only, or for all samples, as discussed below). Thus, especially as there are far fewer proteomic, as compared to transcriptomic, biomarkers studies in general and even fewer proteomic biomarker studies include young children, this study has the potential to be highly impact to the field.

We thank the reviewer for recognizing the unique and substantial strengths of our study. Below we aim to address the noted critiques.

Critiques and suggested improvements:

Major critiques

- The argument for this study would benefit from being more flushed out and better referenced. Although pediatric biomarkers are a major gap in the field, this is not entirely clear from what is written. The introduction would benefit from a deeper literature review addressing the lack of discovery studies done in children and paucity of adult biomarkers that have been studied and shown to be valid in children.

We have added in the introduction further discussion on the lack of biomarkers identified in children or validated from adult studies. We cite a systematic review from Togun et al. that examined biomarkers for childhood TB, and note the wide heterogeneity and overall poor quality of the studies. In addition, we cited a recent review specifically on host blood protein biomarkers from Gaeddert et al., which noted the limited data

for children. We then use these points to further justify the need for a more robust study for children across diverse settings.

- The study would benefit from sub-group analyses, including age <2, sex, HIV status, and nutritional status. For example, there are major differences in HIV status and nutritional status between TB and unlikely TB, and while this is not unexpected, it should be acknowledged and discussed. Furthermore, subgroup analysis may help to determine if these are playing a role in your signature. Finally, Table 1 does not include any statistical analysis of potential differences between cohorts.

We agree that sub analyses of the potential associations between clinical variables and TB status could reveal potential confounding effects and variables to include in biosignature development. We tested the association between the requested clinical variables and TB status by chi-square test. Of all the tested variables, only malnutrition (defined as WAZ < -2 for children under 5 and BMI <18.5 for 5+) and HIV had a significant association with TB status (only looking at Confirmed, Unconfirmed and Unlikely TB). We have added text to the Results and Discussion section to address this.

Variable	chi2	p
sex	0.4103	0.8145
age	4.0877	0.1295
hiv	6.8080	0.0332
malnutrition	8.9365	0.0115

- Did you use a program to analyze your data? If so, where is the code? You refer to a github page about a python package, but there is not code for analysis in python. Did you use any other programs for analysis or graphing?

Yes, we have now included a “Computational packages utilized” section in the methods to provide this detailed information. The text of that section reads as follows:

“Raw proteomics data was analyzed with either MSFragger⁴³ (DDA data) or with DIA-NN (DIA data)⁴⁴, and the generated DDA library and our previous reported plasma library⁴² were merged using easypqp (<https://github.com/grosenberger/easypqp>). For data processing, model training, and figure generation we used the following packages in Python (v 3.8.2): scikit-learn (v1.5.1), pandas (v2.2.2), numpy (v.1.26.4), pyCombat (v), <https://github.com/epigenelabs/pyComBat>), joblib (v.1.4.2), seaborn (0.13.2), matplotlib (v.3.9.2), matplotlib-base (v3.9.2), scipy (v 1.13.1), statsmodel (v 0.14.2). The following packages in R (v.4.3.1, release ‘Beagle Scouts’) were used for figure generation: ggplot2 (v.3.5.1), RcolorBrewer (v1.1.3), viridis (v0.6.5), ggpubr (v0.6.0), ggsci (v3.2.0). Additionally, the GSAR R package (v.1.40.0) was used for analysis of the log2FC between Confirmed and Unlikely TB. All code for data analysis, imputation, and figure plots is available here: https://github.com/anfoss/COMBO_code.git.”

- Statistics used are lacking in some places. There is no methods section on statistics and some components are missing as to what test was used. For example, what statistical test did you use for your volcano plot in Figure 3C?

Different statistical tests are used for different analyses throughout the manuscript, based on what is most appropriate in a given setting. Thus, we have elected to include statistical information within its relevant methods section, rather than as a standalone section. Particularly, in this revision we have included more details in the “DIA-PASEF data analysis” section with regard to statistical tests. We have also included new text

in the legend for Figure 3C to specify that a 2-sided Welch test was used for statistical analysis in the volcano plot.

- The validity of the study is somewhat hard to judge currently, because it is unclear exactly what you have done. The main point of confusion for us was whether or not the model was trained on 75% of the data and then independently validated on 25% of the data? And if so, is the figure 4B related to this and the AUC generated only on the 25% validation cohort? If so, this should be made clear. If not, then we are concerned about the need for validation of the signatures found.

We apologize for the confusion. The reviewer is correct in their interpretation that the model was trained on 75% of the data and then independently validated on the remaining 25%. We have added the following statement in the “Machine learning for identification of a proteomic biosignature for childhood TB disease” section of the methods to make this more clear:

“The remaining 25% of data was then used as a test set for model evaluation.”

Additionally, the reviewer is also correct that figure 4B is the analysis of only the 25% validation cohort. To clarify this we have modified the figure legend for Figure 4B to read as follows:

“B. ROC curves for best-scoring combination of features on the test data (25%).”

- Figure 5 is difficult to interpret. If we assume that unconfirmed TB is mainly truly TB, then the signature did poorly. If, however, we assume the more likely scenario that unconfirmed TB is a mixed picture, then how are we supposed to know your signature is correctly determining the children with TB? Since the children are followed prospectively, is there more information about their long-term outcomes? Is there a different way to validate your data? In addition, regarding Figure 5C, it is not at all clear that there is closer clustering as suggested. Rather, positive in 2/3 of the models appears to be closer than all models. Furthermore, there is significant deviation in the PC2 from all models, that ends up being a similar degree with PC1 for negative unconfirmed. Therefore, there is overstatement of your conclusions here.

We appreciate this comment. All children with Unconfirmed TB were treated for TB, and at follow up 2-3 months later they had improved (this fact is now noted in the manuscript). The reviewer is correct that (1) we assume that the unconfirmed cases represent a mixed population of both children with and without TB, and (2) we cannot know with certainty whether our signature is correctly determining the children with TB among the Unconfirmed TB group, as doing so would require a trial in which anti-TB treatment is provided based on signature results. Alternatively, future studies could consider further phenotyping (e.g., with transcriptomics) to see if these results also support a diagnosis of TB. For now, all we can say is that we were able to identify a “TB signature” in many, but not all children in the Unconfirmed TB group, likely supporting that the Unconfirmed TB group is a mixed picture as noted by the reviewer. We have included text to this effect in the manuscript, particularly to emphasize this uncertainty.

“Moreover, application of these host-biosignatures to children with Unconfirmed TB was able to further support a potential diagnosis of TB in ~63% of cases that were negative by sputum-based testing. Although the lack of microbiological confirmation raises the possibility that these cases did not represent true TB disease, all children had clinical signs and symptoms of TB and improved with anti-TB treatment. However, it is important to note that we cannot know with certainty whether our biosignatures are correct in these classifications of TB among the Unconfirmed TB group. Future clinical trials in which anti-TB treatment is provided based on biosignature results would be required to fully address this question.”

It is also important to note that upon suggestions from Reviewer 5, we have updated our model to be more stringent by allowing only proteins with 50% or less missing values in the Confirmed and Unlikely TB groups to be considered as candidate biomarkers proteins. This has now increased the performance of our model such that 63% of cases (73 out of 115) are similarly classified by all of our biosignatures (3, 4, 5, or 6 protein models) as positive for TB infection within the Unconfirmed group.

Lastly, with regard to figure 5C, the figure interpretation is admittedly not as straightforward as we hoped it would be. We have updated this figure in the manuscript to add ellipses around the different groupings, which we hope will aid in the interpretation. While the reviewer correctly noted the spread on PC2, this axis explains significantly lower variance (25%) than PC1 (37%) meaning that a spread in PC2 is less significant than PC1. As an example, when looking at the density of PC1 for the various classes it is clear that the samples positively predicted by all models are closer to the real distribution of confirmed TB than to the unconfirmed cases that were classified by our biosignature as being negative for TB as demonstrated by both the density plot (A) and the quantile-quantile plot (B).

- The discussion should include comparison with the results of other biomarker studies. If other biomarker studies, particularly other pediatric studies, show similar proteins to your study, this discussion should be added. For example, comparison of the results of this study to the study of a three-marker protein biosignature that distinguishes tuberculosis from other respiratory diseases in Gambian children should be discussed (Togan et al, EBioMedicine, 2020 Aug;58:102909). We cannot see if any of the genes from this study are differentially expressed in your study. If so, how did this model do in your cohorts? Does it predict similar children to have TB in the unconfirmed TB group? If these proteins are not found in your data, why might it be different?

We have added the following text to the Discussion section on this topic:

While this work represents the first untargeted discovery-proteomics biosignature for childhood TB, there have been several cytokine-based signatures identified for TB in children, which would generally be present below the limit of detection with mass spectrometry. However, these targeted analyses were completed at a single center with a small sample size. For example, prior work identified a 3-cytokine signature to distinguish children with TB disease from other respiratory diseases in the Gambia, but they achieved a lower AUC of 0.74 and 72.2% sensitivity.

Additionally, the study suggested by the reviewer demonstrates the use of cytokine arrays to identify a three protein biosignature for pediatric tuberculosis. The three proposed biomarkers (IL-1ra, IL-7 and IP-10), were

not identified. It is notable to mention that cytokines are low in abundance and poorly suited for detection using mass spectrometry based approaches in unenriched samples (PMID: 28951502). Other proteomics-based tuberculosis studies in an adult cohort (PMID: 38512356 and PMID: 32780727) utilize a significantly smaller discovery cohort (10-20 patients vs >100 in our study), where depth in protein detection was prioritized by the use of longer chromatographic gradients and sample preparation approaches. This makes the results of those studies hard to compare. Specifically, previous work utilized extensive off-line fractionation and isobaric labelling which increases exponentially sample processing time while decreasing reproducibility and quantitative accuracy as reported by several groups (PMID: 20382981). As an example, other work have proposed proteins such as CRP and SAA1 (PMID: 32780727), which we (Fig. 3A) and others (PMID 35451001) have demonstrated having no predictive power in separating TB vs non-TB disease, and rather are non-specific inflammatory markers.

- The references need significant work. For example, reference 23 cited in the discussion, is not included in the references section. Reference 22 in the methods does not appear to be the same reference as reference 22 in the discussion. Reference 19 in the discussion refers to a WHO reference but would not be an appropriate reference for IFN γ production by immune cells during Mtb infection yet is also listed here. Reference 18 is out of order, appearing later than reference 19-23 in the text. There are also missing references including statistics that are unreferenced but should be - e.g. "96% of deaths in children are in those whom treatment is not yet initiated" and "...estimated half of the children with TB diseases....".

We appreciate the reviewers attention to detail and apologize for our omissions and formatting errors. We have now included the appropriate references where needed, and fixed missing and out-of-order references.

Minor critiques

- Regarding enrollment into the cohorts, a workflow of how many children were evaluated and included/excluded would add clarity. Also, specifics of why children were excluded should be added.

This was not a prospective evaluation, but based on analysis of banked samples previously collected by the study sites in a 1:1:2 ratio of Confirmed:Unconfirmed:Unlikely TB. We have clarified in this the methods, and added our sampling approach. We have also noted in the discussion that further prospective validation is needed.

- Regarding Supplemental Table 1, it seems there is missing data for some of the donors from The Gambia: C337, C371 and C389-C400. All healthy and latent TB donors are from Uganda. This should be mentioned in the text.

Thanks for catching this issue in Supplementary Table 1. We have corrected the file. Additionally, we added the following text 'we have also included a small number (n = 27) of asymptomatic healthy children from Uganda'.

- Some components that should only be in the methods appear in the results and are somewhat distracting. As examples, the quantity of plasma, the filter-based processing in 96 well plates, and the type of mass spectrometry machine used are appropriately included only in the methods section.

Thank you for this perspective. We have revised that sentence of the manuscript to read as follows:

"For all children, we started from 1 μ L of undepleted plasma and performed high-throughput proteomics sample preparation followed by data-independent acquisition (DIA-PASEF) mass spectrometry analysis (Fig. 1A)."

- Regarding Figure 1, does the data shown in Figure 1B represent the total number of peptides and proteins, or is this the average for each individual? For the 7 outlier samples that were removed, how different were the peptide and protein numbers in these samples? Is there a standard number that would usually be used as a cutoff for this kind of analysis? Are these 7 included in figure C/D? Figure 1F is difficult to follow. Have you looked for proteins that are greater than or less than 4 orders of magnitude from the reference levels? And if so, what is the range of reference levels? What would be the expected distribution of these proteins in an

otherwise healthy population? Finally, “over >4 orders” is redundant. Should be just over 4 or > 4. What is SERPINF2? Is this the lowest abundance protein?

Figure 1B represents the total numbers across all samples. We have modified the legend text to read as follows to clarify this point: “**B. Barplot showing the total number of unique peptide sequences and protein groups identified across all samples.**”

With regard to the 7 outliers that were removed, these samples had substantially reduced numbers of peptides detected. As shown in Figure 1C, we observe an average of 2,628 peptides detected per sample, however for these 7 samples that were removed as outliers there were fewer than 1000 peptides detected in each of these samples and they are not displayed in this figure panel. There is not a standard number of minimum peptides/proteins that would be used to remove, as this can depend highly on a variety of experimental variables. For this reason, we used a criteria of 3 standard deviations as the cutoff for determining outliers. This criteria retains ~99% of the data while removing clear outliers. We have added the following text to the methods section to more fully describe this:

“Samples were excluded if the number of peptides was below 3 standard deviations of the median number of peptides (2591), which removes samples with less than 1700 peptides.”

Lastly, in regard to Fig. 1F. It appears there is confusion about what is represented in the plot. This plot shows the distribution of concentrations for plasma proteins. These concentration values are split up by those that were detected in our proteomics studies (yellow) and those that were not detected (purple). As reported in countless plasma proteomics studies, there is a bias for detection of proteins present at high concentrations, with many proteins of low concentration being below the limit of detection. To help clarify this, we have modified the manuscript text to read as follows: *“While those detections were biased towards proteins of higher concentration, we were able to reproducibly detect proteins down to a level of 12.1 ng/L concentration (SERPINF2), with a median concentration of 40 ng/L (Fig. 1f).”* As well as modified the figure legend text.

Overall, the proteins that are detected in our study span >4 orders of magnitude in concentration values. This range of concentration values detected is not a factor of the cohort population (healthy, age range etc.), but is rather a reflection of the general dynamic range of mass spectrometry detection for undepleted plasma. With regard to “over >4 orders”, this is not redundant, as the “over” reflects that we are describing a range and the “>” indicates that this range spans more than 4 orders of magnitude. For clarity we have modified the text to read as follows: *“spanning more than 4 orders”*.

Lastly, the reviewer is correct in that SERPINF2 is the protein with the lowest concentration that we detect following our strict data processing and filtering. It is notable that prior to preprocessing we identified proteins with even lower reported concentration (GUCY1B1, 5.2 ng/L and PSMD6 8.6 ng/L).

• Regarding Figure 3C, what is meant by manual curation of their localization?

Here “manual curation” was meant to say that we reviewed the literature for evidence that these proteins could be secreted, rather than relying solely on gene ontology terms. We appreciate that this was a confusing description and have now modified it to be described as *“literature evaluation of their localization...”*

• Regarding Supplementary figure 2, what is TopN vs BestN?

We utilized both all combinations of N features as well as the most important features derived from LASSO importance. Accordingly we defined as BestN the feature combination achieving the highest sensitivity at 70% specificity on the test data (25%, not used for training) while as TopN the TopN feature derived from LASSO importance. In other words, top2 are the two most important features from LASSO importance (i.e. absolute magnitude of the model coefficients) when evaluated individually, while best2 refers to the combination of two features achieving the highest evaluation metric.

- You may consider adding to the discussion that the lack of differences between countries may actually validate doing single-country studies.

This is an interesting idea to consider, though this would be bolstered if we had sufficient sample size in the test set to perform subgroup analysis by country.

- The discussion would benefit from the addition of a discussion of limitations.

We have added a limitations section to the discussion.

- Regarding references, #10-12 appear in the text before #9. “WARS1 . . . linked to TB infection via multiple mechanisms” should be referenced.

We appreciate the reviewers attention to detail. We have corrected these reference ordering issues, and have added additional references as suggested.

Reviewer expertise:

- Senior reviewer – expertise in development of pediatric TB biomarkers and in pediatric infectious diseases including tuberculosis.
- Junior reviewer – expertise in TB and in analytic methods.

Reviewer #4 (Remarks to the Author):

Thank you for your time in reviewing our manuscript.

Reviewer #5 (Remarks to the Author):

The paper reports interesting, original and important results in the field of paediatric tuberculosis, and features a good sample size and rational approach to discover a protein signature for sputum-free diagnosis. The inclusion of 4 countries and symptomatic controls is a great strength, as pointed out by the authors.

The paper is well-written and communicates its message concisely.

My main concerns are with the reporting of the data analysis. Below, please see requests for clarification and more detailed reporting of the analysis methods. I recommend the authors use their own discretion whether to add the requested details to the main or supplementary text.

The imputation strategy is clearly explained, but no motivation for it is provided. I am not experienced in proteomics, so it may be that this strategy is standard in the field. It is not clear to me why different imputation strategies were used for values missing in more, or less than 50% of samples. It may also not be clear to other readers. Please provide the rationale for these strategies.

From a proteomics point of view, a peptide can be not identified for a variety of reasons. Three common scenarios include:

1. The peptide (and it's corresponding protein) is truly absent in the sample

2. The peptide (and its corresponding protein) is present at levels which are below the limit of instrument detection
3. The peptide is present above the detection limit, but is not identified due to technical issues such as co-interfering peptides, isobaric peptides, FDR estimation issues, etc.

When a peptide is consistently detected in most samples (>50%) we assume that the lack of identification of that peptide in some samples may be due to scenario 3 above. Here we impute the abundance of that sample in the missing samples by using the mean values in samples where the peptide was detected.

Conversely, when a peptide is sparsely detected (<50% of samples), we assume that a peptide is not detected due to either scenario 1 or 2 above. Here, we want to impute with a small value that represents this assumption of the protein being absent or of low abundance in the samples where it was not detected.

Lastly, in cases of extremely sparse detection (<10% of samples), we assume that this peptide is not of sufficient abundance to be quantitatively reliable, and thus these peptides are excluded.

Please provide a clearer explanation of how the 75% and 25% split data was used. It is clear that the 75% was used for the feature selection with LASSO. But after that, for the all-combinations models, how were the final 6 models selected? Was every one of them applied to the 25% dataset and the final 6 were the best ones? Or were they somehow ranked by performance on the 75% with cross-validation and then only the best 6 were applied to the 25%?

Following LASSO feature selection, we utilized the same 75% to train all models, and the same 25% for testing all models. Each combination model was tested on this 25% (unseen and not used at any training step for any model LASSO or logistic) and the specificity at 90% sensitivity was used as final rank for a specific N combination (i.e. all 2 features models were ranked by the specificity and the best performing one was kept as BestN model for that N=2). Combinations reaching the same sensitivity at 90% sensitivity were ranked by AUC and the one achieving the highest AUC was reported.

We added the following text for clarity: *“The remaining 25% of data was then used as a test set for model evaluation for all models and was not utilized for training at any step in this initial analysis”*

If every single model of the thousands of combinations was applied to the 25% dataset, this has to be made very clear in the Methods and discussed as a limitation in the discussion. The function of a data split is to provide an “untouched” dataset to evaluate overfitting of trained models. If the untouched dataset is utilized for every single trained model, it essentially becomes another training step and not an independent evaluation of the model. The results should be treated with more caution in this case, as an independent evaluation step to confirm the performance, is still lacking.

See response above.

Please make clear for all the results, whether it is all data, the 75% or the 25% result that is reported in every case.

See response above.

How was the training performed? Please provide specifics on the type of cross-validation and the software used. It is only stated that scikit-learn was used for the LASSO step.

For training we utilized `LogisticRegression(penalty='l2')` with default parameters in scikit-learn v1.5.1.

For the combinatorial analysis, it is stated that linear models were used with different combinations of proteins. Linear regression models do not output a classification score, so it is not clear how linear models could have

been used in this step? It is possible that generalized linear models were used with the binomial family and logit link? Please provide details of the model that was used, the software and function, and provide a generic model formula to clarify.

The reviewer is correct and we erroneously noted linear instead of logistic regression, and have corrected this.

Finally, please report the original percentage missingness for each of the markers included in the final models, as this will also influence the confidence in the performance.

We thank the reviewer for the suggested analysis as it allowed us to uncover two proteins (IGLL5 and IGLV3-9) which have high levels of missingness (>70%) across both TB classes and countries (panel A-B below). When comparing the intensity values without imputations using Mann-Whitney U-test we did not identify a significant difference between TB classes, while this was identified for WARS1 (the protein with the third highest degree of missingness in our biosignature) as shown in panel C below. It is important to note that WARS1 has lower average abundance compared to the IGLs abundance, which suggests that the missingness for IGLL5 and IGLV3-9 proteins might be attributed by one or both of the following factors: (1) our initial assumption of low abundance of these proteins across a substantial proportion of patients, and/or (2) individual sequence variation in these specific proteins that prohibit their reproducible detection by mass spectrometry where peptides are mapped to a common reference sequence.

For example, for IGLL5 647 mutations (encompassing ~97% of the 214 amino acid sequence) are reported across the entire sequence and our identification (highlighted in dark red in the plot below) is limited to the peptides with the lowest number of mutations. The combinatorial space of these mutations (203! for a single mutation site across the entire sequence) makes the identification of these protein variants challenging/impossible in our unbiased DIA-MS setup due to FDR instability. The accepted solution to this problem in the field is to utilize a consensus sequence (which we also utilized), but we acknowledge that this misses entirely the individual variability in hypervariable proteins and is an important limitation of this study (and all mass spec based biomarker studies) when analyzing immunoglobulins sequences or proteins with high degree of polymorphism.

To overcome this, we decided to increase the stringency for our initial set of proteins in the biosignature development by selecting only proteins with 50% or less missing values in Confirmed and Unlikely TB (tested together) prior to LASSO feature selection. We then selected from the remaining proteins, combinations exceeding the required WHO target product profile for a diagnostic test. In this analysis we identified two new biosignatures for 5 and 6 proteins exceeding the WHO target product profile. As 5 proteins biosignature we propose IGHV3-33 (20% missing), APOM (0% missing), CD44 (1.5% missing), TNC (21.5% missing) and MMP2 (21% missing) which achieves 93% sensitivity at 70% specificity (0.73-0.97 95%CI, AUC=0.87) and as 6 protein biosignature we propose IGKV1D-33 (0% missing), APOM, CD44, TNC, MMP2 and FCGR3A (23% missing) which achieves 96% sensitivity at 70% specificity (0.78-0.978, 95%CI, AUC=0.88). We updated the entirety of figures 4 and 5 to reflect these changes. We further added to the Discussion that caution should be used when proposing IGGs as potential host markers due to the high polymorphism of these proteins in human populations and this needs to be confirmed by low degree of missingness across the data.

REVIEWERS' COMMENTS

(responses are in blue text)

Reviewer #5 (Remarks to the Author):

Thank you to the authors for the detailed feedback on the previous review comments. Most comments were addressed and the analysis methods have been made clearer in the paper. The authors are also commended for now providing the analysis code in a GitHub repo.

One outstanding issue is the fact that every single model that was trained, was also tested on the test set. It is more robust to train models, then apply them through cross-validation, still only on the training set (or a hold-out set, separate from the test set). Then rank them according to their cross-validation performance on the training set, then select the top 3, or 5 which are then the only ones applied to the test set. In your case, you could have selected the top 2 cross-validated models from every N markers and only applied those to the test set.

The test set serves as protection against overfitting and multiple testing. By applying every single model to the test set, it has basically now become part of the training, and your results are essentially 4-fold cross-validation results. Because of random variation, you would expect 5% false positives, and from millions of potential models, that becomes a huge number. I request the authors to acknowledge this in the limitations section - because every trained model was applied to the test set, the final model results are essentially cross-validation results and have not been conclusively validated on an independent cohort. This does not take away anything from the many strengths of this study, most notably the good sample size and geographic diversity.

We thank the reviewer for this important observation. We agree that evaluating a large number of models on the test set, as was done in our analysis, can introduce optimistic bias due to implicit multiple testing. Ideally, model selection would be performed entirely within the training set (e.g., via cross-validation), and only a small number of top-performing models would then be evaluated on the independent test set to ensure strict separation between model development and validation.

While we did not use the test set during training, we acknowledge that by applying all candidate models to the test set for performance evaluation, it no longer serves as a fully independent hold-out. As such, the reported test performance should be interpreted with this caveat in mind.

We have now included the following statement in the limitations section of the manuscript (*'Finally, our biosignatures were evaluated on the test set and not a fully independent hold-out set, hence the reported performance may be optimistic due to multiple testing, and should be interpreted as exploratory rather than confirmatory.'*) to clarify this point.

The authors may also consider discussing why only logistic regression was attempted for these classification models. Tree-based methods and non-linear methods could also have been evaluated and may have achieved better AUCs.

We evaluated the use of non-linear methods at the beginning of the study (NaiveBayes, SVM, RF, Adaboost, XGBoost and various bagging/stacking/pasting classifiers). We observed similar performance across the board for these models with logistic regression performing better on smaller subsets of features (data not shown). This desirable feature coupled with the inherent better interpretability of logistic regression vs 'black box' models (such as XGBoost, RF, etc) lead us to utilize logistic regression as the final model of choice.

Another useful addition could be to show correlations between all the markers included in all the final models. It may explain why some markers appear in only one model and could aid in taking these markers forward into a different technology. The authors may use their discretion to include this.

We appreciate this suggestion, but have decided not to include this.